# TemDeep: A Self-Supervised Framework for Temporal Downscaling of Atmospheric Fields at Arbitrary Time Resolutions

**Liwen Wang[1], Qian Li[12], Qi Lv[12], Xuan Peng[12] and Wei You[1]**

[1]College of Meteorology and Oceanography, National University of Defense Technology, Changsha, China.

[2]High Impact Weather Key Laboratory of CMA, Changsha, China.

*Correspondence to*: Qian Li (public_liqian@163.com)

**Abstract**

Numerical forecast products with high temporal resolution are crucial tools in atmospheric studies, allowing for accurate identification of rapid transitions and subtle changes that may be missed by lower-resolution data. However, the acquisition of high-resolution data is limited due to excessive computational demands and substantial storage needs in numerical models. Current deep learning methods for statistical downscaling still require massive ground truth with high temporal resolution for model training. In this paper, we present a self-supervised framework for downscaling atmospheric variables at arbitrary time resolutions by imposing a temporal coherence constraint. Firstly, we construct an encoder-decoder structured temporal downscaling network, and then pretrain this downscaling network on a subset of data that exhibit rapid transitions and is filtered out based on a composite index. Subsequently, this pretrained network is utilized to downscale the fields from adjacent time periods and generate the field at the middle time point. By leveraging the temporal coherence inherent in meteorological variables, the network is further trained based on the difference between the generated field and the actual middle field. To track the evolving trends in meteorological system movements, a flow estimation module is designed to assist with generating interpolated fields. Results show that our method can accurately recover evolution details superior to other methods, reaching 53.7% in the restoration rate on the test set. In addition, to avoid generating abnormal values and guide the model out of local optima, two regularization terms are integrated into the loss function to enforce spatial and temporal continuity, which further improves the performance by 7.6%.

## 1 Introduction

In the field of meteorology, temporal downscaling refers to the enrichment of time-series data by filling in the time gaps in observations or numerical products, which can provide a more continuous and comprehensive understanding of geophysical phenomena. Temporal downscaling in atmospheric fields holds considerable importance, given its extensive applications across a wide range of domains. In climate research, accurate temporal interpolation plays a vital role in understanding long-term climate variations and assessing the impacts of climate change (Hawkins & Sutton, 2011; Michel et al., 2021; Papalexiou et al., 2018). By enriching historical climate records with temporally enhanced data, researchers gain a more detailed depiction of past climatic events (Barboza et al., 2022; Neukom et al., 2019). For example, the analysis of high-resolution data has revealed the relationship between global temperature rise and the frequency and intensity of extreme weather events, such as heatwaves and heavy rainfall (Kajbaf et al., 2022; Seneviratne et al., 2012). In the field of weather forecasting, accurate temporal downscaling significantly enhances the quality of short-term weather predictions (McGovern et al., 2017; Requena et al., 2021). Filling gaps between discrete atmospheric observations allows for accurate tracking and prediction of various meteorological phenomena (Dong et al., 2013). For instance, the ability to capture rapid changes in wind patterns using high-resolution temporal data enables more accurate forecasting of severe storms, hurricanes, and their paths. This information is critical for issuing timely warnings, facilitating evacuations, and minimizing the potential damage caused by such weather events (Raymond et al., 2017). Furthermore, high-resolution time-series data aid in optimizing agricultural practices, optimizing energy production from renewable sources, and improving transportation planning by considering detailed weather patterns (Lawrimore et al., 2011; Lobell & Asseng, 2017).

Current methods for temporally downscaling atmospheric fields mainly fall into two categories: dynamical downscaling and statistical downscaling. Starting from a specific initial condition, dynamical downscaling methods can interpolate or extrapolate atmospheric fields to a finer time scale by integrating governing equations over time. Early pioneering work by Lorenz (1963) established the basic framework of using governing equations of fluid

dynamics and thermodynamics to predict future atmospheric states. Since then, models such as the Weather Research
and Forecasting (WRF) model (Skamarock et al., 2008) and the Community Earth System Model (CESM) (Hurrell et
al., 2013) have been developed, incorporating advanced physical parameterizations and data assimilation techniques.
These models have been widely used in producing high-temporal-resolution datasets, such as the European Centre for
Medium-Range Weather Forecasts' Integrated Forecast System (ECMWF IFS) updates (Bauer et al., 2015) and the
High-Resolution Rapid Refresh (HRRR) forecasts (Benjamin et al., 2016). However, the computational expense of
these models is a significant barrier, especially for high-resolution, long-term, or global-scale studies (Maraun, 2010).
In addition, these models require highly accurate initial conditions. Studies by Lorenz (1969) and Palmer et al. (2005)
demonstrate how uncertainties in initial conditions and model parameters can lead to significant prediction errors over
time, referred to as the 'butterfly effect'.
The limitations of dynamical downscaling methods have prompted research into statistical alternatives, as
they are computationally less expensive and can be easily applied across different spatial and temporal scales (Fowler
et al., 2007). These methods, often employing regression techniques or machine learning algorithms, aim to identify
and exploit statistical relationships between low-resolution and high-resolution data, such as weather generators
(Gutmann et al., 2011; Lee et al., 2012), naïve (Chen et al., 2011; Jia-hong, 2006) and autocorrelation (Mendes &
Marengo, 2010). However, as discussed by Maraun (2010), these methods often assume linear or local relationships
in consecutive fields and may oversimplify complex atmospheric dynamics.
In recent years, deep learning has been widely applied to meteorology for its potentials to extract complex
patterns from large amounts of data (Reichstein et al., 2019). For example, Kajbaf et al. (2022) conducted temporal
downscaling with artificial neural networks on precipitation time-series with a 3-hour time step. However, deep
learning applications in meteorology so far have generally relied on supervised learning, requiring large amounts of
high-resolution ground truth data for training, which could be difficult to acquire due to limited observation intervals,
excessive computational demands and high cost of data restoration (Bolton and Zanna, 2019).
In summary, although advancements have been made in temporal downscaling, there still exist significant
demands for methods that can provide high temporal resolution with better physical consistency, improved
computational efficiency and most importantly, less reliance on high-resolution ground truth data. This motivates our
study, which aims to explore self-supervised learning as a potential solution to these challenges. As a form of
unsupervised learning, self-supervised learning is a machine learning method that does not rely on supervision but
leverages supervisory signals from the structure or properties inherent in data to train deep neural networks (Liu et al.,
2020). This approach can leverage vast amounts of unlabeled data for training, thereby significantly enhancing the
model's generalization capabilities. It has been applied in diverse fields, including meteorology science (Eldele et al.,
2022; Pang et al., 2022; Wang et al., 2022).
In this paper, we present TemDeep, the first self-supervised framework for downscaling atmospheric fields
at arbitrary Temporal resolutions based on Deep learning. This framework addresses this issue by imposing a temporal
coherence constraint across time-series fields, which means multiple consecutive fields themselves are leveraged as
supervision information to train the model. Firstly, we construct an encoder-decoder structured temporal downscaling
network, which is capable of performing interpolation at any resolution (see Section 3.6), and pretrain this downscaling
network by designing a composite index to filter out a subset of data with rapid changes (see Section 3.3). The
pretraining stage allows the model to initially capture general patterns and features present in the atmospheric data. In
the next step, we utilize this pretrained model to downscale the fields from adjacent time periods and subsequently
infer the field at the middle time point (see Section 3.4). Then, the model is further trained based on the difference
between the inferred field and the actual middle field, according to the temporal coherence inherent in atmospheric
variables. To effectively track the evolving trends in meteorological system movements, the network adopts a flow
estimation module to assist with synthesizing fields. We have also designed a module to process terrain data, which
enables the model to better perceive the prior information of the underlying surface. In experiments, our method
demonstrates effectiveness in accurate downscaling various atmospheric variables at different temporal resolutions,
reaching over 53.7% in the restoration rate, superior to other existing unsupervised methods.
The structure of this paper is as follows: Section 3 presents the details of the study area and data sources used
in our study. Further, we explain our methodology, specifically detailing the entire training process and network
architecture. In Section 4, we conduct extensive experiments to assess the model's effectiveness. Finally, Section 5
summarizes the methods and contributions made in this study and points out possible future works and applications.

**2 Related Work**

2.1 Temporal downscaling

Time-series downscaling aims to enhance the temporal resolution of a given dataset, a process particularly relevant to meteorology and climate science, where high-frequency observations or model outputs are often needed to capture rapid atmospheric processes. In classical approaches, dynamical downscaling has been extensively explored: running high-resolution numerical weather prediction models (e.g., WRF, CESM) from coarser initial fields (Skamarock et al., 2008; Hurrell et al., 2013). Although this method accounts for complex physical processes, it often incurs prohibitive computational costs, especially for large domains and long simulations (Maraun, 2010). Consequently, statistical downscaling has emerged as a more computationally tractable alternative. Early methods include regression-based techniques that link coarse-scale predictors (e.g., large-scale geopotential height fields) to fine-scale variables of interest (Fowler et al., 2007). However, such methods typically assume linear or semi-parametric relationships, which may be insufficient for capturing non-linear and non-stationary climate signals. Similarly, approaches grounded in stochastic weather generators (Gutmann et al., 2011; Lee et al., 2012) or parametric assumptions (Chen et al., 2011) can fail to represent abrupt changes in meteorological variables, thereby producing over-smoothed time series.

With the rise of machine learning, more sophisticated models for time-series downscaling have surfaced. Supervised deep learning methods—such as feed-forward networks or LSTM-based architectures—have been employed to predict higher-temporal-resolution data from coarse inputs (Kajbaf et al., 2022). These methods can outperform simple interpolation techniques (e.g., linear or spline-based) by learning complex temporal patterns. Nonetheless, a consistent challenge remains: supervised approaches demand substantial ground truth at high temporal resolution for training. In many regions and periods, such data are either unavailable or too expensive to generate using dynamical models.

Recent efforts to address these limitations include semi-supervised or weakly supervised frameworks, where partial or noisy high-resolution data are combined with additional constraints or complementary datasets (Bolton & Zanna, 2019). In parallel, optical-flow-based interpolation techniques have been explored for time-series data, especially in computer vision tasks, to estimate pixel- or voxel-wise motion and generate intermediate frames (Reda et al., 2019). While flow-based methods help track spatial shifts of meteorological features, they often rely on small time-step differences or still require some form of high-resolution reference for validation.

Thus, the demand remains for methods that (1) exploit large volumes of low-resolution time-series data, (2) capture non-linear transitions more effectively than simple averaging, and (3) minimize or eliminate dependence on collocated high-frequency labels. This gap motivates the exploration of purely self-supervised strategies for time-series downscaling, leveraging inherent structure in sequential meteorological data.

2.2 Self-Supervised Learning

Self-supervised learning (SSL) has gained prominence in computer vision and natural language processing for its ability to utilize large unlabeled datasets by creating "pretext tasks" that reveal intrinsic data structure (Liu et al., 2020). Well-known image-based approaches such as SimCLR(Chen et al., 2020) and MoCo (He et al., 2020) train encoders by contrasting augmented views of the same image, thereby learning robust feature representations without category labels. Similarly, BYOL (Grill et al., 2020) employs a bootstrapping strategy to learn latent representations through a student–teacher framework, while CPC (Oord et al., 2018) focuses on maximizing mutual information across different parts of a sequence. These methods have proven highly effective for downstream tasks like classification or semantic segmentation, reducing the need for extensive labeled datasets.

In time-series contexts, SSL has likewise emerged as a compelling direction. One line of work relies on contrastive objectives: for example, splitting time-series data into segments and learning to discriminate between "true" temporal neighbors and randomly sampled distractors (Eldele et al., 2022). Other strategies introduce masked reconstruction tasks—analogous to masked language modeling in NLP—to capture both local and global temporal dependencies (Pang et al., 2022). These generic self-supervised approaches have motivated new research in geoscience, where high-quality labeled data are typically sparse or expensive to obtain (Reichstein et al., 2019).

In meteorology, self-supervision has only recently begun to receive attention. For instance, researchers have explored spatiotemporal contrastive learning to classify weather systems (Wang et al., 2022). The advantage is the ability to harness vast archives of reanalysis or satellite data, circumventing the need for comprehensive manual labeling. Despite these advances, most SSL applications in meteorology focus on classification or feature extraction

rather than downscaling. Adapting the paradigm of frame interpolation from the vision domain to atmospheric fields
remains non-trivial, because meteorological variables exhibit domain-specific physical constraints (e.g., hydrostatic
balance, mass conservation, strong orographic influences).
2.2 Self-supervised learning

Self-supervised learning (SSL) has emerged as a powerful paradigm to leverage large-scale datasets without
requiring explicit labeling (Liu et al., 2020). In contrast to fully supervised methods, SSL derives surrogate tasks from
inherent structures within the data—such as spatial coherence in images or temporal consistency in sequential data—
to generate "pseudo labels" for model training. In meteorological applications, SSL is particularly attractive due to
the massive volume and multivariate nature of atmospheric datasets, which often lack the fine-grained annotations
required for supervised learning (Eldele et al., 2022; Pang et al., 2022).

Recent efforts have demonstrated the potential of SSL to capture complex dynamics in atmospheric fields by
creating training objectives aligned with physical principles or temporal continuity (Wang et al., 2022). Such
approaches help learn robust representations that generalize well across space and time, enabling tasks like weather
system classification, anomaly detection, and data downscaling without the prohibitive cost of manually generating
high-resolution labels. Moreover, self-supervision can naturally exploit the continuity and multi-scale variability
characteristic of climate and weather processes, where adjacent temporal or spatial samples offer substantial
information about underlying physics. By systematically constructing self-supervised tasks around these features,
researchers can improve model fidelity and reduce overreliance on synthetic datasets. In essence, SSL paves the way
for scalable and adaptive meteorological models, transforming abundant but under-labeled atmospheric data into
meaningful insights without heavy labeling requirements.

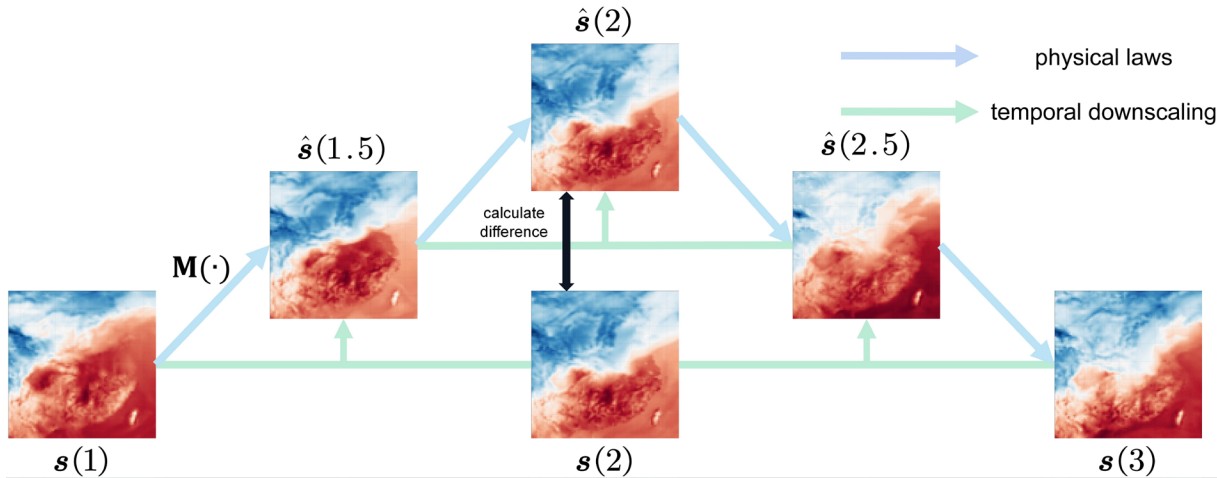

**Figure 1. Illustration of time-evolving atmospheric fields.** The green arrows represent the evolutions of an
atmospheric field guided by physical laws with temporal coherence. The blue arrows represent the outputs of a
temporal downscaling model, which seeks to approximate the physics-guided evolutions.

In fact, although atmospheric variables do not change linearly at different time steps, it is commonly believed
that their evolutions are consistently guided by the same physical laws and thus exhibit temporal coherence over time
(Lorenz, 1969). In other words, for the state $s(t)$ of any atmospheric variable at any time $t$, it will transition from
$s(t-1)$ to $s(t)$ following the mapping $\mathbf{P}$ guided by a set of physical laws, expressed as $\mathbf{P}: s(t-1) \to s(t)$. Based
on this invariant mapping constraint, time-series data themselves can be used as supervision information to train the
deep learning model. To be specific, at any moment $t$, $s(t)$ can be taken as the truth value to train the mapping
relationship from $s(t-1)$ to $s(t)$. As shown in the example in Fig. 1, for three consecutive fields $s(1)$, $s(2)$ and
$s(3)$ with an interval of 1 hour, if the goal is to train a downscaling model $\mathbf{M}$ to fill the gaps at $1.5h$ and $2.5h$ and
obtain $\hat{s}(1.5) = \mathbf{M}(s(1), s(2))$, $\hat{s}(2.5) = \mathbf{M}(s(2), s(3))$, after generating $\hat{s}(2) = \mathbf{M}(\hat{s}(1.5), \hat{s}(2.5))$, the
existing $s(2)$ can serve as supervision and the errors between $\hat{s}(2)$ and $s(2)$ be utilized as loss to train $\mathbf{M}$. Therefore,
it is clear that continuous atmospheric variables inherently contain sufficient information, which can be utilized as
supervision for self-supervised temporal downscaling.

## 3 Data and Methods

### 3.1 Study area and dataset

Our study focuses on the geographic area bounded by longitude 100°E to 125°E and latitude 20°N to 45°N
with a spatial resolution of 0.25°×0.25° (see Fig. 2), and data for this region were downloaded from the European
Centre for Medium-Range Weather Forecasts (ECMWF) ERA5 reanalysis dataset. The dataset comprising 87,660
two-hour-interval samples from 2001 to 2020 is used as the training dataset. The testing dataset consists of 8,760 one-
hour-interval samples in 2021. To evaluate the generalization performance of the TemDeep method, experiments were
conducted on three atmospheric variables: 2-meter air temperature (*t2m*), 850*hPa* geopotential height (*z*) and 850*hPa*
relative humidity (*rh*). Horizontal and vertical wind components are utilized to calculate wind speed as part of a
composite index (see Section 3.3). Recognizing the influence of topography on local climate and weather patterns, we
have also included terrain data with a resolution of 15*km*, sourced from NASA's Shuttle Radar Topography Mission
(Hennig et al., 2001). This resolution is sufficient to represent major terrain influences on atmospheric processes
without significantly increasing computational demands.

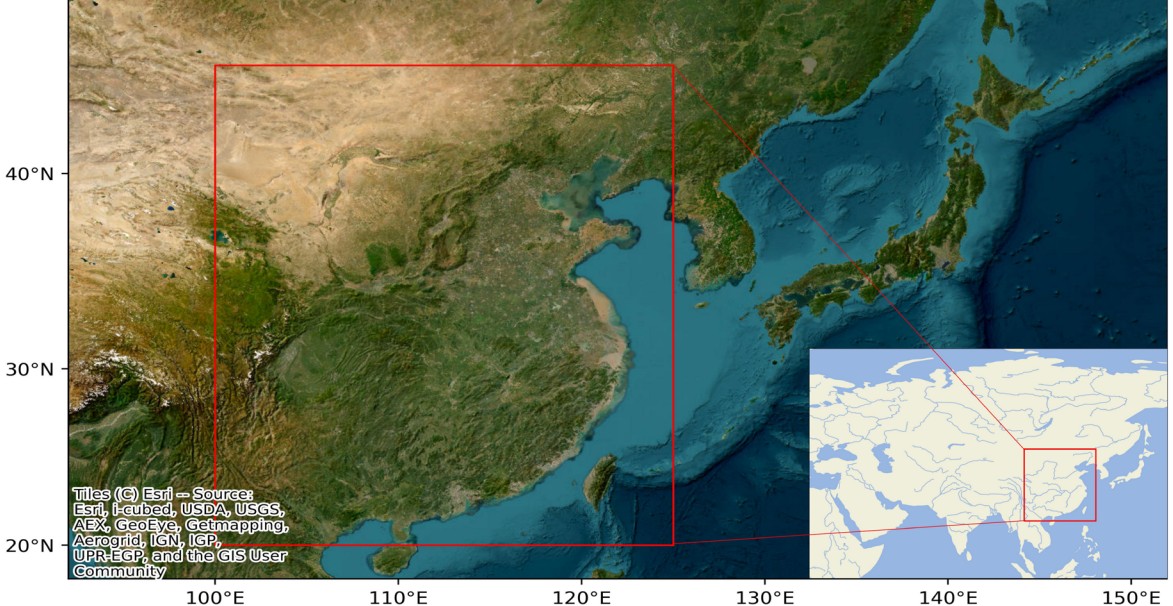

**Figure 2. Satellite image of the study area.** The study area is outlined by the red rectangle (base map imagery
provided by Esri WorldImagery).

### 3.2 Problem definition and overview

Given the initial atmospheric fields $\{\boldsymbol{p}\}_T\left(\boldsymbol{p}\in\mathbb{R}^{n_x\times n_y}\right)$ represented as a continuous gridded dataset with a
temporal resolution of $\tau$, our goal is to achieve temporal downscaling at any resolution $\theta\tau\left(\theta\in(0,1),\mathbf{R}^+\right)$. Here,
$n_x$ and $n_y$ denote the number of grid points in the horizontal and vertical directions, respectively. That is, for a given
period of weather process occurring between the interval $[T_0, T_0+\tau]$, we aim to accurately generate the interpolated
field at any time point $T_0+\theta\tau$. To achieve this goal, a self-supervised framework is presented for temporal
downscaling (see Fig. 3), in which the training procedure consists of two primary stages. In the first stage, we pretrain
our model on a subset of data to simulate the training process on a real high-resolution dataset by selecting scenarios
with rapid transitions. Then, the model is further trained under guidance of a temporal coherence constraint, leveraging
supervision information inherent in the low-temporal-resolution time-series. In addition, two regularization terms are
utilized in the loss function to guide the model out of local optima and prevent abnormal values.

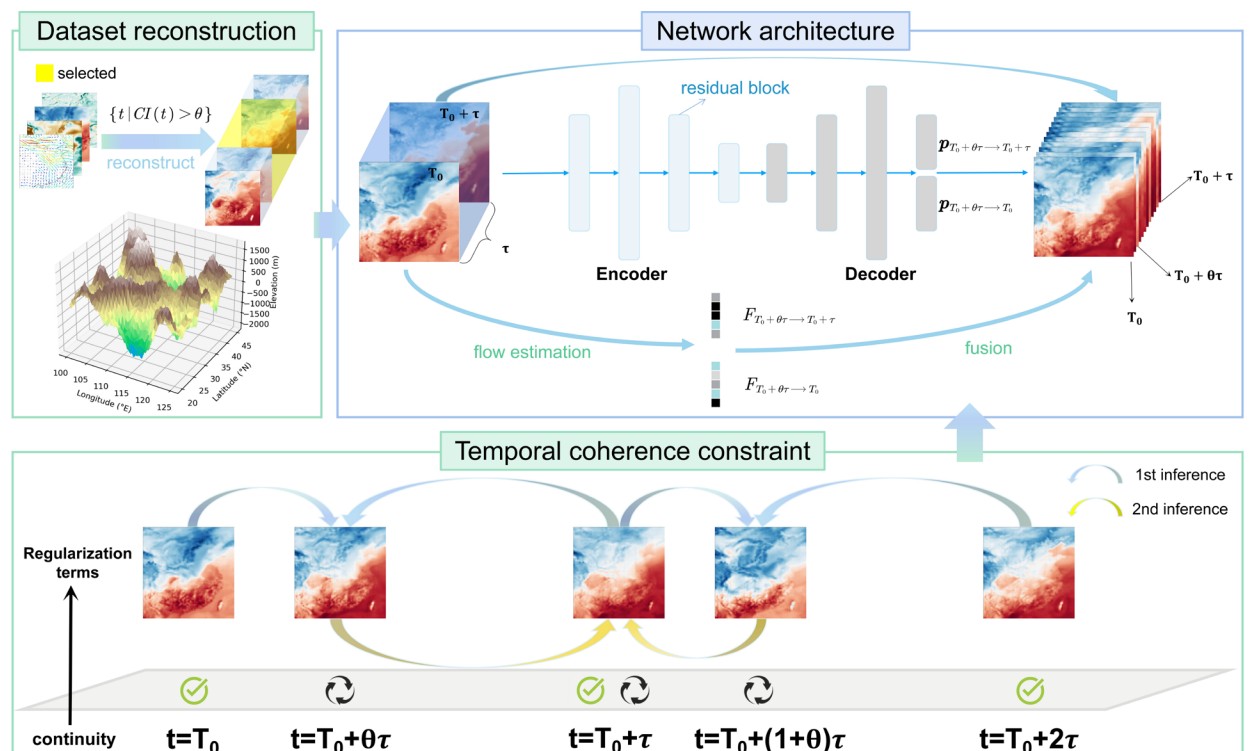

**Figure 3. Overview of the proposed TemDeep framework for self-supervised temporal downscaling.** The overall network structure for temporal downscaling is depicted in the top right portion of the figure, which is composed of an encoder-decoder structured field prediction network and a flow estimation module, taking consecutive fields and terrain data as input.

### 3.3 Reconstructing a pretraining dataset through self-similarity

It is easily understood that scenarios with rapid transitions could reflect a condensed evolution of atmospheric processes, where changes that might typically occur over longer durations are instead experienced in a compressed time period(Davis, 1994; Stanley, 1997). Therefore, these scenarios occurring within shorter time intervals in low-temporal-resolution data can potentially serve as 'pseudo labels' for scenarios within longer time intervals in high-temporal-resolution data.

Based on this kind of self-similarity across time scales, we propose to reconstruct a pretraining dataset by establishing a composite index to filter out scenarios with rapid transitions. This composite index is designed based on four physical variables that are indicative of weather system transformations, respectively *rh*, *t2m*, 850*hPa* wind speed (*v*) and 850*hPa* vertical velocity (*w*). Rapid changes in wind speed can indicate major weather phenomena, and similarly, humidity changes are key to atmospheric stability and sudden shifts can trigger severe convection. *t2m* gradients drive atmospheric circulation, with steep gradients signifying developing weather fronts. Lastly, vertical velocity indicates vertical air movement and can signal cloud formation or precipitation.

Given an atmospheric variable $a$, the normalized change for each time step $t$ is defined as $\Delta a(t) = (a(t+1) - a(t))/\sigma(a)$, where $\sigma(a)$ represents the standard deviation of the variable $a$ over the entire period. Let $V(t) = [\Delta rh(t), \Delta t2m(t), v(t), w(t)]$ denote the vector of normalized changes, and $W = [w1, w2, w3, w4]$ be the respective weight vector, the composite index $CI$ can be expressed as

$$CI(t) = W^T \cdot V(t) + \eta \sum_{i<j} (w_{ij} \cdot V_i(t) \cdot V_j(t)). \tag{1}$$

Here, the superscript $T$ denotes vector transposition, and the summation extends over all unique pairs of variables $(i < j)$. The parameter $\eta$ is a scaling factor set at 0.02, which can be adjusted to regulate the influence of the

interaction terms. $w_{ij}$ represents the weights linked with the interaction terms, ensuring that each variable has the
same magnitude before multiplication. The first component $W^T \cdot V(t)$ is a linear combination of normalized changes
to quantify individual influence of each variable, while $\sum_{i<j}(w_{ij} \cdot V_i(t) \cdot V_j(t))$ is introduced to account for
synergistic effects among variables by measuring the product of changes between pairs of variables. Finally, we
empirically set a threshold $\theta$ for the composite index at 0.75, and scenarios with a $CI$ value above the threshold are
considered to exhibit rapid transitions:
$$\mathbf{T} = \{t \mid CI(t) > \theta\}, \tag{2}$$

where $\mathbf{T}$ denotes scenarios with rapid transitions. Finally, we obtained a collection of 1,391 scenarios with 12,531
consecutive fields and group these samples every 3 fields into 4,177 sets. During the pretraining process, we train the
model by providing the model with the two adjacent fields as input and tasking it to generate a result that is close to
the middle field in the sequence.
3.4 Self-supervised training leveraging temporal coherence
In our approach, we propose a self-supervised training process, which leverages temporal coherence within
continuous atmospheric fields to generate interpolated fields at arbitrary time resolutions. Taking inspiration from the
success of unpaired data to data translation in a variety of fields (Gao et al., 2022; Reda et al., 2019; Zhou et al., 2016;
Zhu et al., 2017), we define a time-domain temporal coherence constraint, ensuring that the interpolated data point
$\hat{\boldsymbol{p}}_{T_0+\tau}$ created at time $T_0+\tau$ right between $\boldsymbol{p}_{T_0}$ and $\boldsymbol{p}_{T_0+2\tau}$ must consist with the original middle data point $\boldsymbol{p}_{T_0+\tau}$.
That is, as illustrated in Fig. 3, for a given triplet of consecutive data fields, we generate two intermediate data points
in the first inference: one between the first two data points $\hat{\boldsymbol{p}}_{T_0+\theta\tau} = \mathbf{M}(\boldsymbol{p}_{T_0}, \boldsymbol{p}_{T_0+\tau}, \theta\tau)$, where $\mathbf{M}$ is our downscaling
network (see Section 3.6), and the other between the last two data points $\hat{\boldsymbol{p}}_{T_0+(1+\theta)\tau} = \mathbf{M}(\boldsymbol{p}_{T_0+\tau}, \boldsymbol{p}_{T_0+2\tau}, \theta\tau)$. Then in
the second inference, we generate an interpolated data point between these newly created intermediate data points,
$\hat{\boldsymbol{p}}_{T_0+\tau} = \mathbf{M}(\hat{\boldsymbol{p}}_{T_0+\theta\tau}, \hat{\boldsymbol{p}}_{T_0+(1+\theta)\tau}, (1-\theta)\tau)$. In this case, $\hat{\boldsymbol{p}}_{T_0+\tau}$ should match the original middle input data point $\boldsymbol{p}_{T_0+\tau}$,
illustrating the concept of temporal coherence. By changing the time parameter $t = \theta\tau(\theta \in (0,1))$, our method is
capable of generating an array of interpolated data points that maintain temporal coherence over time, effectively
enriching the temporal resolution of the atmospheric dataset. To enforce temporal coherence, we aim to minimize the
difference between $\hat{\boldsymbol{p}}_{T_0+\tau}$ and $\boldsymbol{p}_{T_0+\tau}$, expressed as $\arg\min_{\Phi(M)} \|\hat{\boldsymbol{p}}_{T_0+\tau} - \boldsymbol{p}_{T_0+\tau}\|_1$, then the coherence loss $L_c(\Phi)$ can be
defined in the form of $L_1$ loss:
$$\mathcal{L}_c(\Phi) = \|\hat{\boldsymbol{p}}_{T_0+\tau} - \boldsymbol{p}_{T_0+\tau}\|_1. \tag{3}$$

3.5 Spatio-temporal continuity regularization
Despite the application of the temporal coherence constraint to train the model, which allows for the
simulation of evolving weather systems, it is still necessary to regulate the model to prevent it from local optima and
avoid generating abnormal values. To address this concern, our approach leverages the inherent continuity of
atmospheric fields in space and time, which is integrated into our model training process as a regularization term in
the loss function. An example of spatial and temporal gradients in *t2m* fields is provided in Fig. 4, and Fig. 5 indicates
that 99.59% of the horizontal gradients and 99.31% of the vertical gradients are lower than $3K$ respectively. Similarly,
in the continuously varying fields, 99.55% of the temporal gradients are lower than $3K$. Therefore, it can be assumed
that the majority of grid points in *t2m* fields exhibit strong spatial and temporal continuity, as well as other densely
distributed atmospheric variables, such as geopotential height and relative humidity. Here, spatial continuity implies
that nearby locations should share similar atmospheric conditions, and our model incorporates a spatial continuity loss
term to ensure smoothness in both horizontal and vertical directions:
$$\mathcal{L}_s = \frac{1}{2}\left( \sum_{(x,y) \subset \Omega} \|\hat{z}(t, x+1, y) - \hat{z}(t, x, y)\|_1 + \sum_{(x,y) \subset \Omega} \|\hat{z}(t, x, y+1) - \hat{z}(t, x, y)\|_1 \right), \tag{4}$$

where $\hat{z}(t,x,y)$ represents the model's prediction at time $t$ and location $(x,y)$. Meanwhile, temporal continuity
assumes that the atmospheric conditions do not change abruptly over short periods, and accordingly, our loss function
includes a temporal continuity term that penalizes substantial differences between the model's predictions at three
consecutive time steps:

$$\mathcal{L}_t = \lambda \left( \| \hat{z}(t+1) - \hat{z}(t) \|_1 + \| \hat{z}(t) - \hat{z}(t-1) \|_1 \right), \qquad (5)$$

where $\hat{z}(t)$ denotes the model's prediction at time $t$, and $\lambda$ is a parameter set at 0.35 to control the weight of
temporal continuity in the loss function.

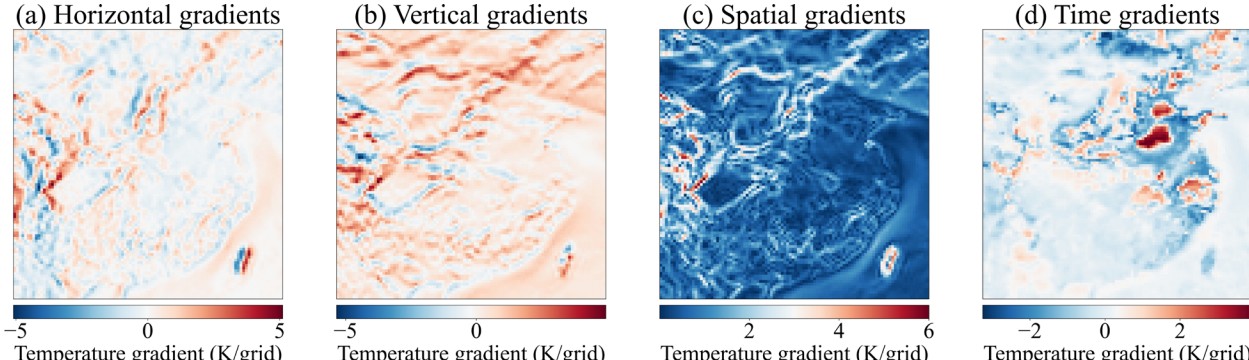

**Figure 4. Spatial and temporal gradients in *t2m* fields at 2021-01-01 08:00:00.** (a) Horizontal gradients computed
along the x-direction, (b) Vertical gradients computed along the y-direction, (c) Spatial gradients showing the
magnitude of the combined horizontal and vertical temperature derivatives, and (d) Time gradients illustrating the
change in *t2m* relative to the preceding timestep. The color scale (in K) indicates where temperature varies most
rapidly: red denotes warming (positive gradients) and blue denotes cooling (negative gradients). Notably, strong
gradients in panels (a)–(c) often align with complex terrain features, highlighting topographic influences. Meanwhile,
temporal gradients in (d) capture abrupt weather changes between consecutive timesteps, underscoring the dynamic
evolution of near-surface atmospheric conditions.

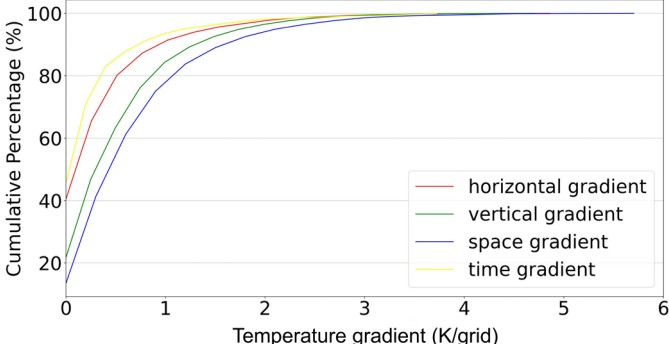

**Figure 5. Cumulative percentage of spatial and temporal gradients.** The x-axis represents the gradient magnitude
(in K per grid or K per timestep), and the y-axis denotes the cumulative percentage of grid cells or timesteps below
that gradient threshold. Each curve corresponds to one type of gradient: horizontal (red), vertical (blue), combined
spatial (green), and temporal (yellow). The plot reveals that the vast majority of temperature gradients in both space
and time are relatively small (e.g., below 3 K), while only a small fraction exhibits larger gradients.
3.6 Network architecture
In this section, we will introduce the network architecture of TemDeep for generating interpolated fields. As
illustrated in Fig. 3, the field prediction network, serving as the backbone network, adopts an encoder-decoder
structure to generate intermediate fields (see Fig. 6, Section 3.6.1). Meanwhile, the flow estimation module adopts a
unique combination of larger convolutional kernels and Leaky ReLU activations to capture long-range motions (see
Fig. 7, Section 3.6.2). Finally, intermediate fields and estimated flow are fused to synthesize interpolated fields
(Section 3.6.3).
3.6.1 Field prediction network

The field prediction network is composed of an encoder-decoder architecture with the inclusion of residual
blocks(Azad et al., 2024), as shown in Fig. 6. It takes consecutive single-element fields and terrain data as input, and
outputs intermediate fields. The encoder part includes four primary components, each comprised of a convolutional
layer and a subsequent residual block. These convolutional layers, coupled with ReLU activation functions, process
input data through multiple filter sizes (64, 128, 256, and 512 filters respectively). Notably, the first convolutional
layer incorporates a 7×7 kernel with a stride of 2 and padding of 3, enabling more robust feature extraction at the
initial stage, while subsequent layers employ 3×3 kernels with a stride of 1 and padding of 1. After each convolutional
layer, a corresponding residual block follows, with in-channels and out-channels matching the corresponding
convolutional layer's filter size. These residual blocks consist of two convolutional layers and ReLU activation
functions, which helps in preserving the identity function and facilitates deeper model learning without the problem
of vanishing gradients. The decoder part is designed to upsample and reconstruct the encoded field back to its original
resolution. It consists of four deconvolutional layers, each applying the ConvTranspose2d function for upsampling,
and these layers upsample the data from 512 filters back to 2 filters, which corresponds to the output flow. Notably,
the kernel size used in these layers is 4 with a stride of 2 and padding of 1, which efficiently enlarges the spatial
dimensions back to the original size. After a convolutional layer, we obtain forward and backward prediction results:
$\vec{p}$ and $\overleftarrow{p}$.

Additionally, to process topographic information and integrate it into input, we introduce a Convolutional
Terrain Integration Module (CTIM). The CTIM employs a convolutional layer with 3×3 kernels, to create an
intermediate feature map topographic information. Subsequent to the convolution operation, batch normalization is
applied to accelerate the training process, followed by a ReLU activation function to introduce non-linearity. This
output then passes through a second convolutional layer with 3×3 kernels to further refine the feature representation.
Once again, we apply batch normalization and ReLU activation to this output. The resulting output from the CTIM is
a set of terrain feature maps, ready to be fed into the prediction network.

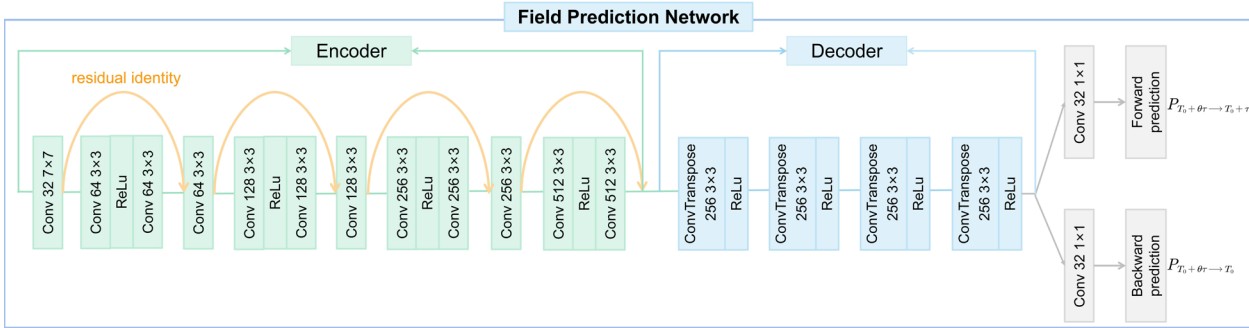


**Figure 6. Detailed architecture of the field prediction network.** The encoder consists of four convolutional layers,
each followed by a residual block (shown by the orange arcs). These layers progressively expand the number of feature
channels from 32 up to 512 through increasing filter sizes (e.g., 32 → 64 → 128 → 256 → 512), while ReLU
activations introduce non-linearity. The decoder mirrors this process with four deconvolution (ConvTranspose) layers
to restore the spatial resolution and reduce the channel depth, ultimately yielding two separate output branches
3.6.2 Flow estimation module

The flow estimation module aims to estimate motion information and calculate forward and backward flow,
which is then fused with the intermediate fields from the field prediction network to assist with generating interpolated
fields. Fig. 8 provides an example of calculated flow in *t2m* fields.

The flow encoder is structured similarly to the encoder of the field prediction network, which comprises four
convolutional layers, each followed by a Leaky ReLU activation function. The initial layer utilizes a 7 × 7
convolutional kernel to extract features from the input, stepping down to a stride of 2 and padding of 3. Following
this, the subsequent layers use 3×3 convolutional kernels with a stride of 1 and padding of 1, moving from 64 to 128,
256, and finally to 512 filters for a more detailed and intricate feature extraction. The subspace features obtained at
this layer, after undergoing convolution and ReLU, yield an activation map $\vec{V}$ (see Eq. 6).
The flow decoder includes five deconvolution layers that upscale the downsampled encoder outputs. Each
layer employs a bilinear upsampling technique to double the spatial dimension, followed by two convolutional layers
and a Leaky ReLU activation. Finally, we obtain forward flow $\mathcal{F}_{T_0+\theta\tau \rightarrow T_0+\tau}$ and backward flow $\mathcal{F}_{T_0+\theta\tau \rightarrow T_0}$ after
two convolutional layers.

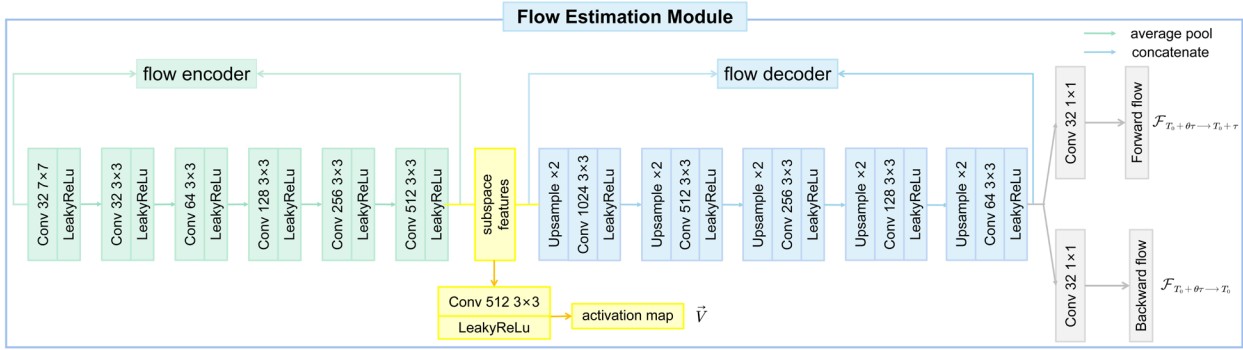


**Figure 7. Detailed architecture of the flow estimation module.** The flow encoder applies successive convolutions
(7×7 followed by 3×3 kernels) and LeakyReLU activations to extract progressively deeper motion features from input
fields. After reaching 512 filters, a final 3×3 convolution and LeakyReLU produce a subspace activation map. In the
flow decoder, multiple upsampling stages (e.g., by a factor of 2) and 3×3 convolution layers with LeakyReLU
progressively restore spatial resolution, eventually yielding two 1×1 convolutions that predict forward flow and
backward flow .

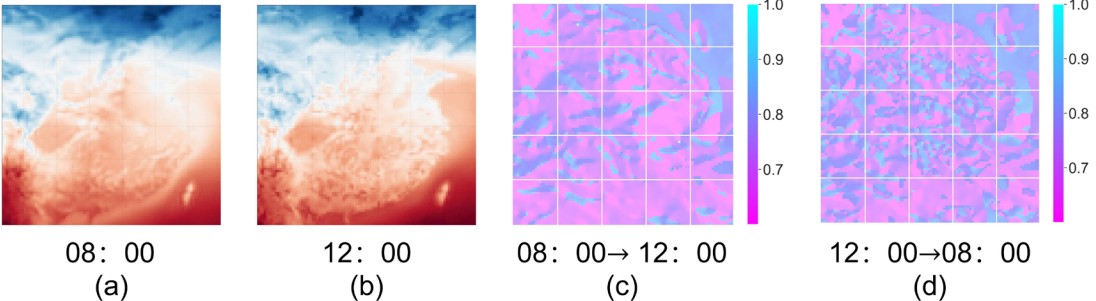


**Figure 8. Forward and backward flow visualization. A** and **b** represent the *t2m* fields at 08:00 and 12:00 on January
1, 2021, while **c** and **d** represent the forward flow from 08:00 to 12:00 and backward flow from 12:00 to 08:00,
respectively.
3.6.3 Fusion and loss function
We can synthesize the target field $\hat{\boldsymbol{p}}_{T_0+\theta\tau}$ by fusing the outputs from the field prediction network and the
flow estimation module as follows:

$$\hat{\boldsymbol{p}}_{T_0+\theta\tau} = (1-\theta)\overleftarrow{V} \odot g\left(\overleftarrow{\boldsymbol{p}}, \mathcal{F}_{T_0+\theta\tau \rightarrow T_0}\right) + \theta\vec{V} \odot g\left(\vec{\boldsymbol{p}}, \mathcal{F}_{T_0+\theta\tau \rightarrow T_0+\tau}\right), \qquad (6)$$

where $g(\cdot)$ is a warping function (Jiang et al., 2017). $\vec{V}$ represents the activation map, referring to whether pixels
remain activated when moving forward from $t = T_0$ to $t = T_0+\theta\tau$ and $\overleftarrow{V}$ is calculated by $\overleftarrow{V} = \left[1 - \vec{V}(i,j)\right]$.

In order to make the estimated flow resemble the actual flow, we utilize it as motion information to further assist in enhancing the quality of field reconstruction, and accordingly, the flow estimation loss can be defined as

$$\mathcal{L}_{\mathcal{F}} = \left\| \boldsymbol{p}_{T_0} - g\left(\bar{\boldsymbol{p}}, \mathcal{F}_{T_0 + \theta\tau \to T_0}\right) \right\|_1 + \left\| \boldsymbol{p}_{T_0 + \tau} - g\left(\vec{\boldsymbol{p}}, \mathcal{F}_{T_0 + \theta\tau \to T_0 + \theta\tau}\right) \right\|_1 \tag{7}$$

Finally, the loss function to train the model can be expressed by combing the coherence loss $\mathcal{L}_c$ (Section 3.3), flow estimation loss $\mathcal{L}_{\mathcal{F}}$ (Section 3.5.3), and continuity loss $\mathcal{L}_s + \mathcal{L}_t$ (Section 3.4) as

$$\mathcal{L} = \mathcal{L}_c + \mathcal{L}_{\mathcal{F}} + \alpha(\mathcal{L}_s + \mathcal{L}_t), \tag{8}$$

where $\alpha$ is a parameter set at 0.35 to adjust the weight of continuity regularization.

3.7 Evaluation metrics

In order to evaluate the performance of our model, we propose three metrics: restoration rate ($Re$), consistency degree ($CS$), and continuity degree ($CT$). Among them, $Re$ is primarily utilized for the evaluation of the discrepancy between the downscaled results and the true values, while $CS$ and $CT$ are auxiliary metrics for the analysis and comparison of different methods.

The restoration rate measures the degree to which our model recovers lost information compared to simple linear interpolation, and a larger $Re$ indicates a better downscaling performance. Let the restoration rate of linear interpolation as zero, then the formula for calculating $Re$ is as follows:

$$Re = 1 - \frac{\frac{1}{\Omega} \sum_{(x,y) \subset \Omega} |D^{truth}(x,y) - D(x,y)|^2}{\frac{1}{\Omega} \sum_{(x,y) \subset \Omega} |D^{truth}(x,y) - D^{lin}(x,y)|^2}. \tag{9}$$

In this formula, $D^{truth}$ is the ground truth, $D$ are the data generated by our model, $D^{lin}$ is calculated through linear interpolation and $\Omega$ represents all pixels in the field.

The consistency degree is a metric used to evaluate the level of consistency in generated fields, and a larger $CS$ indicates a smaller discrepancy between the estimated flow $\hat{F}_{T_0 \to T_0 + \theta\tau}$ and the true flow $F_{T_0 \to T_0 + \theta\tau}$. It is calculated based on the flow estimation module and can be expressed as

$$CS = 1 - \frac{\left\| \mathcal{F}_{T_0 \to T_0 + \theta\tau} - \hat{\mathcal{F}}_{T_0 \to T_0 + \theta\tau} \right\|_1}{\left\| \mathcal{F}_{T_0 \to T_0 + \theta\tau} \right\|_1}. \tag{10}$$

The continuity degree measures how smoothly the preceding field transitions to the next, and a larger $CT$ indicates more smoothness. The mathematical representation is

$$CT = 1 - \frac{\left| \left\| p_{T_0 + \theta\tau} - p_{T_0} \right\|_1 - \left\| \hat{p}_{T_0 + \theta\tau} - p_{T_0} \right\|_1 \right|}{\left\| p_{T_0 + \theta\tau} - p_{T_0} \right\|_1}, \tag{11}$$

where $\hat{p}_{T_0 + \theta\tau}$ is the interpolated field and $p_{T_0}$ is the preceding field.

**Table 1. Performance comparison among different methods based on *Re*.** According to Eq. 9, the result of linear interpolation is set to 0 as the basis for comparing other methods. Among all unsupervised comparison methods, TemDeep achieved the best performance, approaching the supervised TemDeep\*. *rh* and *z* are only downscaled from 2-hourly to 1-hourly intervals.

| Methods | *t2m* (2h→1) | *t2m* (3h→1) | *t2m* (4h→1) | *t2m* (5h→1) | *z* (2h→1) | *rh* (2h→1) |
|---|---|---|---|---|---|---|
| Linear | 0.000 | 0.000 | 0.000 | 0.000 | 0.000 | 0.000 |
| Cubic spline | 0.102 | 0.041 | 0.019 | 0.018 | 0.135 | 0.074 |
| Optical flow | 0.219 | 0.188 | 0.102 | 0.059 | 0.342 | 0.236 |

| | | | | | | |
|---|---|---|---|---|---|---|
| non-flow | 0.462 | 0.431 | 0.359 | 0.307 | 0.505 | 0.417 |
| non-regular | 0.499 | 0.470 | 0.397 | 0.325 | 0.525 | 0.488 |
| non-pretrain | 0.528 | 0.501 | 0.433 | 0.372 | 0.568 | 0.489 |
| TemDeep | *0.537* | *0.508* | *0.442* | *0.376* | *0.576* | *0.498* |
| TemDeep* | **0.682** | **0.641** | **0.579** | **0.430** | **0.701** | **0.553** |

Note: "non-flow," "non-regular," and "non-pretrain" represent methods that do not include flow estimation, regularization modules, and pretraining steps, respectively.

## 4 Results and discussion

We conduct experiments on an Ubuntu 20.04 system equipped with eight A100 GPUs. The TemDeep model is trained using the adam optimizer (Kingma & Ba, 2014) with an initial learning rate of 1e-5, and a mini-batch size of 256. Downscaling results of *t2m*, *z* and *rh* fields at different time resolutions, respectively 2, 3, 4, 5 and 6 hours, into 1-hour time intervals, are shown in Table 1.

4.1 Quantitative analysis

In order to evaluate the effectiveness of our proposed method on temporal downscaling, we select several methods that do not require supervision information for comparison, namely linear interpolation, cubic spline interpolation and optical flow-based interpolation. The linear interpolation method computes the average value between adjacent fields, while cubic spline interpolation, using four data fields, achieves a smooth curve with cubic polynomials. Additionally, optical flow-based interpolation estimates pixel motion between fields to predict their state at a desired time point. As illustrated in Table 1, for the six tasks of *t2m* (2h→1), *t2m* (3h→1), *t2m* (4h→1), *t2m* (5h →1), *z* (2h→1), and *rh* (2h→1), the TemDeep method scores 0.537, 0.508, 0.442, 0.376, 0.576, and 0.498 in *Re*, respectively, all considerably higher than the scores achieved by other methods under unsupervised conditions. The approximate time for each inference is 600ms. Without the pretraining stage, *Re* is relatively lower on all tasks, suggesting that this stage is important in initially capturing general patterns in atmospheric data. The supervised training condition TemDeep* method scores the highest, implying that supervised training can further enhance the downscaling performance of the TemDeep method.

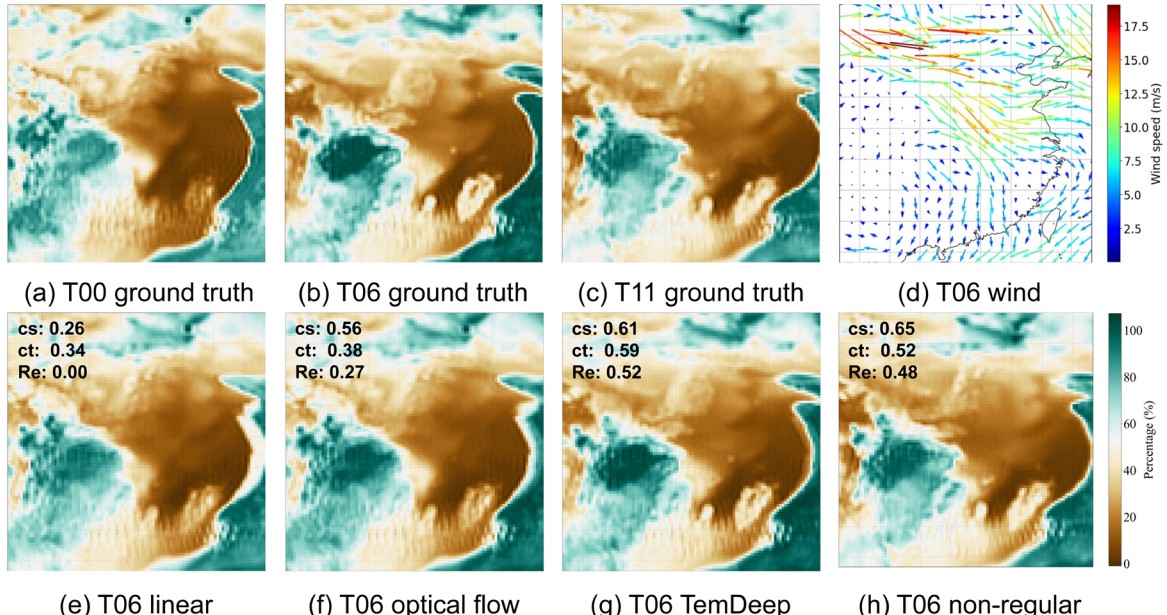

(a) T00 ground truth    (b) T06 ground truth    (c) T11 ground truth    (d) T06 wind

(e) T06 linear    (f) T06 optical flow    (g) T06 TemDeep    (h) T06 non-regular

**Figure 9. Visualized comparison. a**, **b**, and **c** respectively represent the true values of *rh* fields on January 1, 2021. **d** represents the wind direction and speed at 850 hPa at 06:00, where the wind in the central region points towards the

southeast, driving the dry air mass in the same direction, resulting in the expansion of the dry area towards the southeast.
**e**, **f**, and **g** display the interpolation results at 06:00 obtained through different methods. **h** represents the result from
TemDeep when the spatio-temporal continuity regularization is removed.
The flow estimation module provides an improvement of 0.075 in *Re* by guiding the model to learn the
movement of weather systems, and the result demonstrates more consistency with the trends of weather system
movements, as illustrated in Fig. 9g. In contrast, if completely ignoring the motion of weather systems, the result of
time interpolation would simply be an average of the preceding and succeeding fields, leading to significant errors
compared to the ground truth, as shown in Fig. 9e. The spatio-temporal continuity regularization also provides an
improvement of 7.6% from 0.499 to 0.537 in *Re* by ensuring the generated fields be consistent with the observed
patterns in the input data. As depicted in Fig. 9h, without this regularization, the model occasionally produces
erroneous estimates of the intensity and direction of motion. Nevertheless, with the inclusion of the regularization
term, the results are inevitably constrained to linear changes to a certain degree, which has conflicts with the actual
non-linear evolutions.

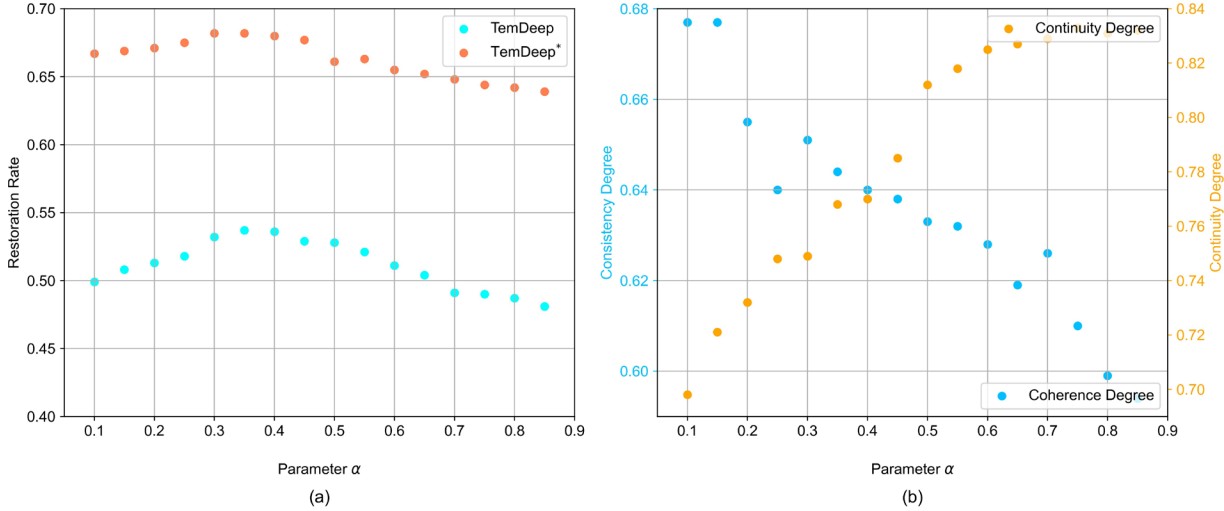


**Figure 10. Model performance under the enforcement of spatio-temporal continuity with varying weights. a**
shows *Re* of TemDeep trained under self-supervised conditions and supervised conditions (denoted as $\mathrm{TemDeep}^*$)
at different $\alpha$. **b** shows consistency degree and continuity degree of TemDeep at different $\alpha$.
To strike a balance between the spatio-temporal continuity regularization and actual non-linear evolutions,
we introduce a parameter $\alpha$ in the loss function to adjust the weight for regularization and conduct ablation studies,
with the results shown in Fig. 10. A larger $\alpha$ implies that the model emphasizes on regularization, and thus CT
increases while CS decreases. Finally, $\alpha$ is set at 0.35 and *Re* reaches a maximum of 0.537.
Fig. 11 shows the restoration rate of the test set in these experiments. Increasing the training dataset size
consistently improves model performance, but the impact diminishes gradually. Once the amount of training data
reaches a critical value (e.g., 8,760), further increases no longer result in significant improvements, suggesting the
model is reaching its performance limits. When the data volume reaches 26,280, doubling the data leads to only a
modest 1-2% improvement.

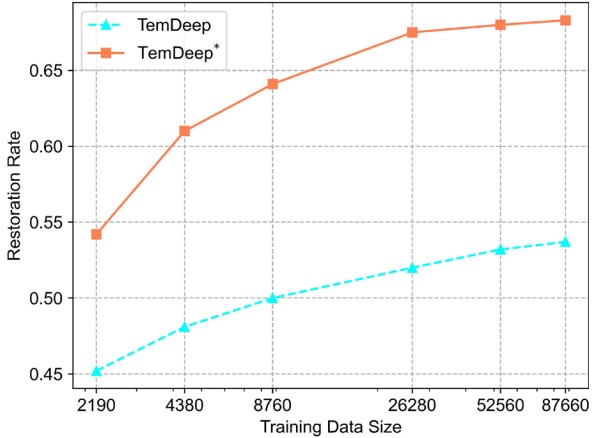


**Figure 11. Restoration rate versus training data size for *t2m* fields.** The x-axis shows the amount of training data (number of 2-hourly samples), while the y-axis indicates the restoration rate, a measure of how effectively the downscaled results recover fine-scale temporal variations. Two methods are compared: TemDeep (self-supervised) and TemDeep* (supervised).

4.2 Case study

In this section, a case study is employed to explore TemDeep's ability in recovering evolving details of *t2m*, *z* and *rh* fields, as shown in Fig. 12. Hourly interpolation is conducted between 08:00 and 12:00 on January 1, 2021, to obtain three interpolated fields at 09:00, 10:00 and 11:00 (UTC).

In the temporal interpolation of *t2m* fields, the selected area in January exhibits a noticeable temperature difference between the sea and the land at 12:00 compared to 08:00, and the gradual changes occurring at 09:00, 10:00, and 11:00 are clearly reproduced by the TemDeep method. Due to the sensitivity of *t2m* to altitude, the temperature gradient near the Sichuan Basin is clearly depicted, closely aligning with the contour of the actual altitude gradient, as marked by the rectangle. Most importantly, at 10:00, regions marked by the triangles exhibit large surrounding gradients and non-linear abrupt changes, resulting in a lower continuity degree of 0.54. In this case, the TemDeep method still achieves a high precision in reproducing the field, with a restoration rate of 0.49, reaching 0.48 and 0.52 at the preceding and following field, respectively.

For the 850hPa *z* fields, their variations are relatively simpler compared to the *t2m* fields, making downscaling easier and leading to less precision fluctuation. The average *Re* over the three-hour period reaches 0.56. At 08:00, there exists a high-pressure region on the western edge, surrounded by low pressure, resulting in a significant gradient. In the generated *z* fields, this gradient gradually diminishes from 09:00 to 11:00, and the central high-pressure region moves northeastward and eventually dissipates, as marked by the ellipse and arrow, which evolves closely in accordance with the ground truth.

Similarly, in the three generated *rh* fields, the drier region on the eastern edge can be observed slowly moving eastward, consistent with the ground truth. At 08:00, the drier region is still located some distance away from the 125°E line, but after four hours of continuous changes, the easternmost part of the dry region has already crossed the 125°E line, and TemDeep has reproduced this movement of dry air mass, rather than simply averaging the fields.

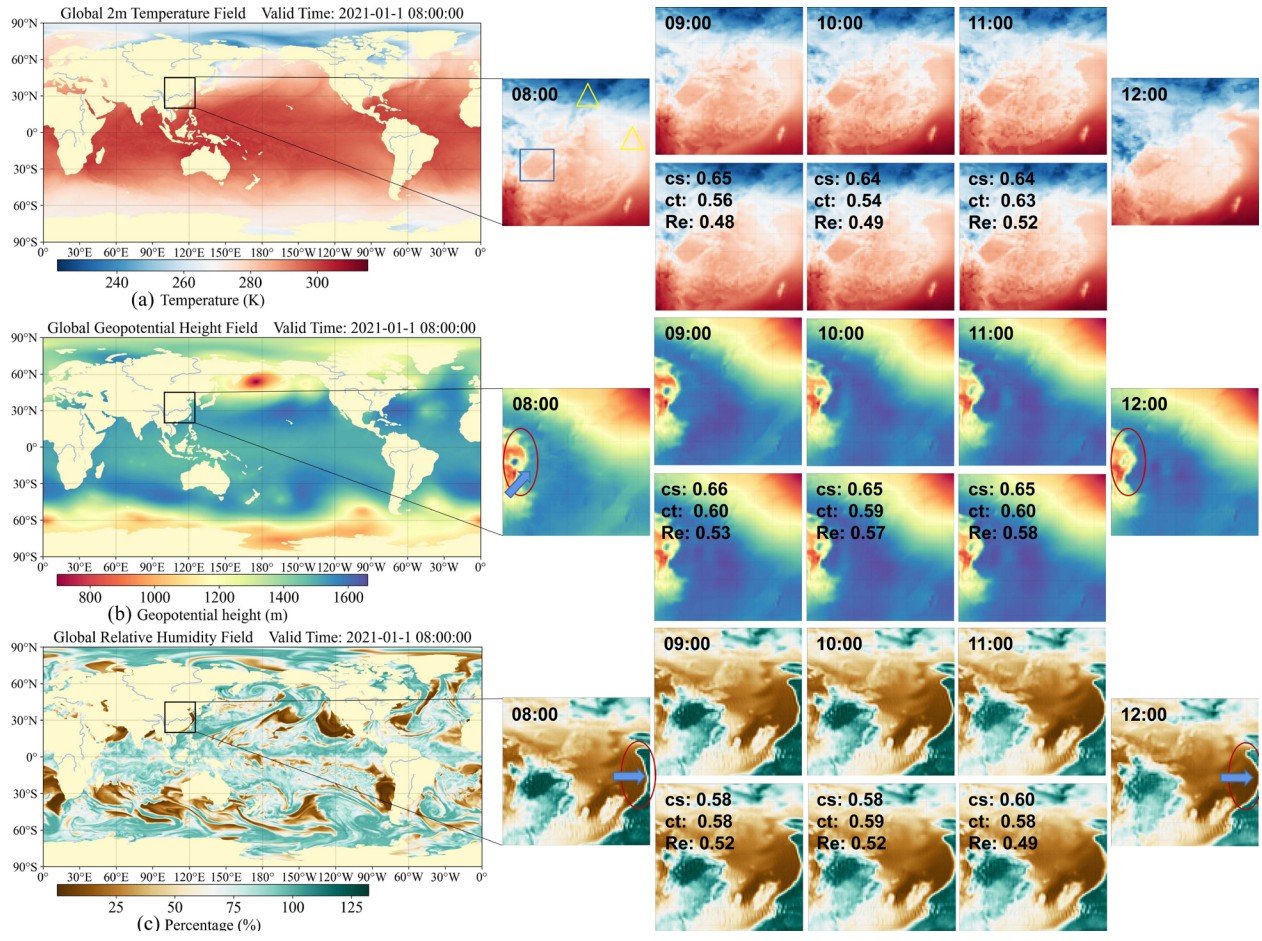

**Figure 12. Hourly downscaling results for t2m, geopotential height, and relative humidity fields from 08:00 to 12:00 on January 1, 2021.** Each row focuses on a different atmospheric variable, with a global map on the left and a zoomed-in region of interest (black box) on the right. The enlarged panels show the interpolated fields at 09:00, 10:00, and 11:00, alongside the actual field at 08:00 and 12:00. Colored shapes (e.g., circles, triangles) highlight notable features such as strong gradients or rapidly shifting weather systems.

4.3 Discussion

This study addresses a persistent challenge in atmospheric science: generating high-resolution temporal data without relying on expensive high-frequency observations. By proposing a self-supervised framework that leverages temporal coherence, our method contributes to the growing body of literature on data-driven downscaling approaches, particularly those aiming to reduce dependence on ground-truth, high-resolution data (Kajbaf et al., 2022; Bolton & Zanna, 2019). Unlike traditional supervised deep learning methods, which require substantial labeled datasets, our approach relies on the inherent temporal dynamics present in consecutive reanalysis fields, thereby extending the notion of self-supervised learning (Liu et al., 2020) to meteorological time series.

A key novelty lies in the pretraining step that extracts "rapid-transition" samples, inspired by the notion that compressed, abrupt changes can serve as effective surrogates for higher-frequency transitions (Davis, 1994). By specifically targeting periods of strong gradients in temperature, humidity, and wind fields, our model can learn the nuanced behavior of evolving weather systems without explicit high-resolution labels. This differs from standard statistical downscaling methods (Chen et al., 2011; Mendes & Marengo, 2010) that typically assume linear or limited autocorrelation structures. Moreover, the incorporation of a flow estimation module to track and warp features aligns with previous literature on optical-flow-based interpolation (Reda et al., 2019), yet we extend these ideas by enforcing additional spatial and temporal continuity. Such continuity constraints provide a safeguard against physically implausible discontinuities—a limitation observed in simpler interpolation or purely optical-flow-based methods (Lorenz, 1963).

Another valuable contribution is the explicit integration of terrain information. While dynamical downscaling methods (Skamarock et al., 2008) naturally handle topographic influences, they often incur high computational costs. Our approach achieves a similar fidelity in representing orographic effects at a fraction of the computational effort. This aligns with recent trends in using auxiliary data (e.g., terrain or land cover) to refine regional climate modeling (Barboza et al., 2022). In doing so, we bridge a gap between purely physics-based models and data-driven approaches by allowing the model to incorporate physical priors in a flexible, trainable manner.

**5 Conclusions**

This paper proposes a self-supervised model for downscaling atmospheric fields at arbitrary time resolutions by leveraging temporal coherence. This model combines an encoder-decoder structured field prediction network with a flow estimation module, fuses intermediate fields and motion information of weather systems and finally synthesizes fields at desired time points. We first pretrain the model based on a reconstructed dataset to initially capture data patterns, and then further utilize existing consecutive fields as supervision for model training. Experiments on three variables ($t2m$, $z$, $rh$) indicate that the proposed TemDeep model can accurately reconstruct the evolutionary process of atmospheric variables at the 1-hourly resolution, superior to other unsupervised methods.

As for future research, we will explore multi-modal data fusion to leverage complementary information from various sources. Since ERA5 only provides data at 1-hour temporal resolution, further research will focus on identifying datasets with higher temporal resolution for more accurate downscaling. Further, we plan to extend our downscaling model based on previous work of self-supervised weather system classification (Wang et al., 2022), that is, to downscale temporal and spatial data by referring to similar types of weather systems through similarity search in the historical dataset. To enable real-time downscaling and more refined forecasting, we will also work on simplifying the model architecture to reduce computational complexity, making it more feasible for deployment in operational environments where fast processing times are critical.

*Code and Data Availability Statement*. All data necessary to reproduce the results of this work can be downloaded at https://doi.org/10.24381/cds.bd0915c6 and https://doi.org/10.24381/cds.adbb2d47. The scripts used for downscaling are freely available at https://github.com/GeoSciLab/TemDeep and have also been permanently archived at Zenodo with the DOI https://zenodo.org/records/14062314.

*Author contributions.* L. W. Wang was primarily responsible for the design of the model and conducting the experiments. Q. Li prepared the experimental datasets and organized the entire research project. Q. Lv, X. Peng, and W. You contributed to the optimization of the experimental code.

*Competing interests.* The contact author has declared that none of the authors have any competing interests.

*Acknowledgments:* This research was funded by the National Natural Science Foundation of China (Grant No. 42075139, U2242201, 42105146), the China Postdoctoral Science Foundation (Grant No. 2017M621700), Hunan Province Natural Science Foundation (Grant No. 2021JC0009, 2021JJ30773, 2023JJ30627)

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
