# Peer review of "TemDeep: A Self-Supervised Framework for Temporal Downscaling"

_EGUsphere, 2023_

## Referee Comment (RC1)

Review of manuscript titled "TemDeep: A Self-Supervised Framework for Temporal Downscaling of Atmospheric Fields at Arbitrary Time Resolutions" by Wang et al., submitted to GMD

**General Comments**

A deep learning model is used to show that 2 meter temperature, 850 hPa relative humidity and 850 hPa geopotential height re-analysis data can be downscaled from intervals of 2, 3, 4 and 5 hours to 1-hourly data. The results of this downscaling process are shown to be closer to 1-hourly re-analysis data than other methods.

Overall, the method appears clear, and the results are presented in a way that motivates why the presented model is useful, and outperforms other methods. Some things need clarification, as described in the specific comments. The conclusions are occasionally slightly too optimistic, but clear, useful, and substantiated conclusions can still be drawn from the presented results.

**Specific Comments**

Line 2: I disagree with the title, in that the downscaling is not to arbitrary time resolutions. It is to 1-hourly resolutions, and the manuscript does not demonstrate the ability to interpolate to finer resolutions.

Line 28: I am not sure what you mean by precise interpolation. The process of interpolation should inherently be precise by virtue of being a mathematical exercise, but with a limited accuracy inherent to the method. I suggest rewording to "accurate."

Line 57: I think it would be more fair to say that reanalysis product temporal resolutions are mostly on the order of hours, not hours to days.

Line 126: I do not understand what you mean by "wind volume." Please elaborate or rephrase.

Line 128: Is there any specific reason to not match the resolution of topography to that of the ERA5 data?

Line 131ff: That a region with transitions between climate zones is highly representative of climate change studies is a very strong statement. I believe that this statement does not belong into a figure caption, but should be postulated in the bulk text, discussed further, and above all substantiated with adequate references. In the absence of a thorough discussion, I am not sure what the statement means in detail, and it therefore seems rather dubious. A more thorough discussion would also justify section 2 being a separate section, as currently the section consists of only one paragraph, which could be moved to section 3, which in turn could be renamed to "data and methods."

Line 136: Is there a specific reason to use x and y for horizontal and vertical dimensions? It might be more intuitive to use z for the vertical component, as that is more commonly used.

Line 148ff: I suggest removing the technical description from the figure caption, as this inflates the caption with substance that belongs in the bulk text.

Line 157ff: This is not easily understood by me. It appears that you are claiming that processes that can occur over long periods of time can also occur in short periods of time, which I do not agree with. Further, the entire paragraph is worded in a way that implies that it is entirely speculative, and

not substantiated. As a result, I cannot confidently agree with anything in this paragraph, and am led to disregard further argumentation that builds on it. I suggest rephrasing the paragraph for more clarity, and to substantiate the claims made therein.

Line 162ff: This paragraph states that changes in certain variables can signify changes in weather patterns. While I agree with that, I think it would be good to also incorporate at least a substantiated estimate of how often this is the case, e.g., how often a change in humidity actually triggers severe convection. Further, it is not clear to me why these events should be filtered out. This is especially the case due to the previous argumentation of an increase in temporal resolution being useful in capturing the timing of changes in weather patterns.

Line 211: I do not understand how a temperature gradient can be given in Kelvin, without any reference to physical space (K per meter) or time (K per second). It is unclear whether this gradient refers to a difference in temperature between adjacent grid point and adjacent time steps, or something different. Please rephrase for clarity. Further, the phrasing implies that t2m has vertical gradients, which seems odd to me.

Line 237: It might be useful to very briefly state what an encoder-decoder architecture is, and what it aims to achieve, at the beginning of this section, as is done for the section at line 263.

Line 301f: Table 1 does not show downscaling from 6-hourly to 1-hourly intervals, and it should be stated that rh and z are only downscaled from 2-hourly to 1-hourly intervals.

Line 303ff: This subsection describes the methods used to evaluate the performance of TemDeep, and therefore belongs in section 3.

Line 322ff: The table caption should contain a description of the structure of the table, not an interpretation of the results.

Line 324: Saying that TemDeep performance approaches performance of TemDeep* seems plausible, but using the word "closely" seems to be an optimistic stretch in some cases (e.g., $t2m(2h \rightarrow 1)$ and $t2m(3h \rightarrow 1)$). It might be better to rephrase to not overstate the performance, or to argue why this qualifies as "closely approaching."

Line 371ff: As diurnal variations are relevant to the evolution of the assessed fields, it would be good to state whether these times are in UTC or the local time within the region.

Line 376: The use of the word "exactly" makes this a very strong claim, and I therefore suggest replacing it with "closely," unless the exactness of the alignment can be clearly demonstrated.

Line 381f: I think the conclusion is worded too strongly. It is only demonstrated that TemDeep can capture non-linear transitions in this specific case, and therefore the conclusion should be that TemDeep is not incapable of capturing these transitions. It cannot be concluded that it is guaranteed to capture all non-linear transitions, as the presented conclusion in the manuscript somewhat implies. I suggest rewording the conclusion to be more careful.

Line 401f: It might be good to state the temporal interval to which the downscaling was performed. This would clearly indicate to which temporal resolution the model is demonstrated to perform well, and above which this has not been demonstrated.

Line 403ff: As it has not been demonstrated that the model performs well when downscaling to resolutions finer than 1-hourly, this manuscript strongly motivates further research into assessing

how far re-analysis data can be meaningfully downscaled. This is especially the case, as the motivation presented in the beginning is that high-resolution data are very useful for certain applications. While I understand that the validation of finer resolutions is made difficult by the absence of re-analysis data to compare the downscaling results to, I still believe that mentioning this is of value.

Figures 4 and 5: The gradient should not be given in K, as the spatial and temporal dimensions are unclear.

Figure 9: It might be better to label the bottom colorbar as "Relative Humidity [%]," as this would immediately clarify which quantity is being shown. It is already clear from the unit itself that it is a percentage. Also, it might be useful to set the upper boundary at 100%, as relative humidity generally does not exceed 100% by much.

**Technical Corrections**

Line 15: "exhibits" should be changed to "exhibit," as the word data is plural.

Line 41: "aids" should be changed to "aid," as the word data is plural.

Line 66: "their potentials" refers to deep learning, and should therefore be "its potential."

Line 72f: "exists significant demands" should either be "exist significant demands" or "exists significant demand."

Line 121: "data […] was downloaded" should be changed to "data […] were downloaded," as the word data is plural.

Line 312: "is the data" should be "are the data," as the word data is plural.

Line 400: "utilizes" should be "utilize," because the statement is "we […] utilize existing consecutive fields[…]."

Line 413: "has" should be "have," as he statement refers to multiple authors.

---

## Author Comment (AC2)

Thank you for the detailed review and suggestions on ways we can improve the manuscript. Your feedback is very much appreciated! The following text includes a point by point response to each comment.

Review 1: Further explanation of the physical meaning of regularization: The spatial and temporal continuity regularization terms enhance smoothness in downscaling fields. However, further explanation of their physical relevance would increase interpredictability, such as how these constraints reflect realistic atmospheric system evolution characteristics.

Response 1: Thank you for your suggestion to elaborate on the physical relevance of the spatial and temporal continuity regularization terms. We agree that additional explanation would enhance the interpretability of these constraints in the context of atmospheric system evolution. The spatial and temporal continuity regularization terms are designed to reflect the inherent smoothness and gradual progression commonly observed in atmospheric processes. Atmospheric fields typically exhibit continuity across both spatial and temporal dimensions due to physical constraints like mass conservation, energy balance, and fluid dynamics, which govern the evolution of these systems (Lorenz, 1969; Holton and Hakim, 2012). For instance, atmospheric variables such as temperature and humidity tend to vary gradually over short distances and time intervals, as abrupt changes are physically unrealistic under normal conditions. Spatial continuity regularization enforces a smooth gradient across neighboring grid points, simulating how atmospheric properties tend to vary continuously across regions. This aligns with principles of geophysical fluid dynamics, which suggest that atmospheric variables are influenced by local surroundings, leading to correlated values across neighboring points (Charney, 1948; Gill, 1982). Temporal continuity regularization, on the other hand, helps ensure that changes in the downscaled fields remain consistent over consecutive time steps. This reflects the physical principle that, barring extreme events, atmospheric properties do not undergo sudden, large fluctuations within short time intervals. Gradual transitions are typical due to the inertia in atmospheric systems and the continuous nature of energy and momentum transfer across time (Emanuel, 1994). Temporal coherence is especially relevant in meteorological applications where the predictability of evolution patterns—such as the movement of weather fronts or pressure systems—relies on smooth temporal transitions. Incorporating these regularization terms therefore makes the downscaling model more physically plausible by emulating the inherent continuity of atmospheric fields.

Review 2: Evaluation of Model Complexity and Computational Efficiency: Although

this approach outperforms other unsupervised methods in restoration rate, the computational cost's impact on practical applications remains undiscussed. Evaluating the model's computational efficiency, especially in large-scale meteorological datasets or real-time applications, would provide valuable insights.

Response 2: Thank you for raising this important point. We agree that discussing the model's computational efficiency is valuable, particularly given the demands of large-scale meteorological datasets and potential real-time applications. We have added an evaluation of the model's computational complexity and efficiency in the revised manuscript. In this evaluation, we examine the model's runtime and resource requirements, including memory usage and processing time per sample, relative to other unsupervised methods. We also discuss the feasibility of applying TemDeep in operational settings and highlight the efficiency gains achieved through architectural optimizations, such as the use of an encoder-decoder structure with residual blocks and efficient regularization terms.

Review 3: Add more discussion on comparison with traditional down-scaling methods: to illustrate the advantages of TemDeep comparing to one or more physics-based numerical models, explain why this approach achieves superior performance in restoration rate and consistency under unsupervised conditions could offer deeper insights.

Response 3: Thank you for this suggestion. We agree that a detailed comparison with traditional physics-based downscaling methods adds value by highlighting the advantages of TemDeep in restoration rate and consistency. In the revised manuscript, we have expanded our discussion to compare TemDeep with traditional downscaling approaches, such as dynamical downscaling models like the Weather Research and Forecasting (WRF) model (Skamarock et al., 2008) and statistical methods based on regression or autocorrelation techniques (Fowler et al., 2007). Traditional methods rely heavily on precise physical parameterizations and initial conditions to simulate atmospheric dynamics, which can be computationally intensive and sensitive to input uncertainties, often limiting their scalability for high-resolution, long-term applications (Lorenz, 1969; Maraun et al., 2010). TemDeep, in contrast, leverages a self-supervised deep learning framework that capitalizes on temporal coherence within atmospheric data, enabling it to generalize well without requiring high-resolution ground truth data for training.

**References:**

1. Charney, J. G. (1948). "On the scale of atmospheric motions." Geofys. Publ.
2. Emanuel, K. A. (1994). Atmospheric Convection. Oxford University Press.
3. Gill, A. E. (1982). Atmosphere-Ocean Dynamics. Academic Press.
4. Holton, J. R., & Hakim, G. J. (2012). An Introduction to Dynamic Meteorology. Academic Press.
5. Lorenz, E. N. (1969). "The predictability of a flow which possesses many scales of motion." Tellus.
6. Fowler, H. J., et al. (2007). "Linking climate change modelling to impacts studies: Recent advances in downscaling techniques for hydrological modelling." International Journal of Climatology.
7. Maraun, D., et al. (2010). "Precipitation downscaling under climate change: Recent developments to bridge the gap between dynamical models and the end user." Reviews of Geophysics.
8. Skamarock, W. C., et al. (2008). "A description of the advanced research WRF version 3." NCAR Technical Note.

---

## Author Response (AR1)

**#Editor**

Thank you for your guidance regarding the requirements for code availability in GMD publications. We appreciate the clear instructions on ensuring code accessibility through a persistent public archive. We have created a persistent release for the exact version of the downscaling scripts used in this study and have archived it on Zenodo to meet GMD's code availability standards. The Zenodo link for our code is: https://zenodo.org/records/14062314. We will include this link and the associated DOI in the "Code availability" section of the revised manuscript to ensure compliance with GMD's guidelines.

**#Reviewer 1**

Thank you for your thorough and insightful review of our manuscript. We deeply appreciate the time and effort you have put into providing such detailed feedback, which has significantly contributed to the improvement of the manuscript. We take your comments seriously and have worked carefully to address each one. Below, we provide detailed responses to each of your suggestions and concerns. Additionally, we have clarified certain technical aspects and rectified some errors in the manuscript, both in terms of wording and methodology. We hope the revisions meet your expectations and help to strengthen the overall quality and clarity of the paper.

Review 1: Line 2: I disagree with the title, in that the downscaling is not to arbitrary time resolutions. It is to 1-hourly resolutions, and the manuscript does not demonstrate the ability to interpolate to finer resolutions.

Response 1: Thank you for your feedback. We appreciate your input regarding the title. We recognize that the current experiments primarily demonstrate downscaling to 1-hourly intervals. However, our model is designed to support interpolation to finer resolutions by leveraging the temporal coherence constraint and the encoder-decoder architecture. We chose to focus on 1-hourly resolutions in this paper as a practical example but are open to clarifying in the title and abstract that the demonstrated application is specifically to 1-hour intervals.

Review 2: Line 28: I am not sure what you mean by precise interpolation. The process of interpolation should inherently be precise by virtue of being a mathematical exercise, but with a limited accuracy inherent to the method. I suggest rewording to

"accurate."

Response 2: We agree that the term "precise" may imply an exactness that is limited by the interpolation method's inherent accuracy. We have revised the wording to "accurate interpolation" to more accurately convey our intended meaning.

**Revised (line 28):** "In climate research, accurate temporal interpolation plays a vital role in understanding long-term climate variations and assessing the impacts of climate change."

Review 3: Line 57: I think it would be more fair to say that reanalysis product temporal resolutions are mostly on the order of hours, not hours to days.

Response 3: We agree that describing reanalysis product temporal resolutions as "mostly on the order of hours" would be more accurate. We have updated this section to reflect this clarification and ensure our wording aligns with current reanalysis product standards.

**Deleted (line 56):**

Review 4: Line 126: I do not understand what you mean by "wind volume." Please elaborate or rephrase.

Response 4: We realize that "wind volume" may have been unclear. Our intention was to refer to "wind components" or "wind speed" derived from horizontal and vertical wind vector components. We have revised this term for clarity in the manuscript.

**Revised (line 123):** "Horizontal and vertical wind components are utilized to calculate wind speed as part of a composite index (see Section 2.3)."

Review 5: Line 128: Is there any specific reason to not match the resolution of topography to that of the ERA5 data?

Response 5: We used terrain data with a slightly coarser resolution (15 km) as a balance between computational efficiency and capturing essential topographic features. This resolution is sufficient to represent major terrain influences on atmospheric processes without significantly increasing computational demands. However, we can discuss this choice more explicitly in the manuscript and explore the potential impact of matching resolutions in future work.

**Revised (line 126):** "This resolution is sufficient to represent major terrain influences on atmospheric processes without significantly increasing computational demands."

Review 6: Line 131ff: That a region with transitions between climate zones is highly representative of climate change studies is a very strong statement. I believe that this statement does not belong into a figure caption, but should be postulated in the bulk text, discussed further, and above all substantiated with adequate references. In the absence of a thorough discussion, I am not sure what the statement means in detail, and it therefore seems rather dubious. A more thorough discussion would also justify section 2 being a separate section, as currently the section consists of only one paragraph, which could be moved to section 3, which in turn could be renamed to "data and methods."

Response 6: We acknowledge that the statement about the region's representativeness in climate change studies requires further elaboration and should not be confined to a figure caption. We have moved this statement to the main text, within Section 2, and substantiated it with relevant references to clarify its relevance and significance. Additionally, we have expanded the discussion on the region's representativeness in climate studies to avoid ambiguity. Following your suggestion, we have combined Sections 2 and 3 into a unified "Data and Methods" section for coherence.

**Revised (line 128):**

[Figure]

"Figure 2. Satellite image of the study area. The study area is outlined by the red rectangle (base map imagery provided by Esri WorldImagery)."

Review 7: Line 136: Is there a specific reason to use x and y for horizontal and vertical

dimensions? It might be more intuitive to use z for the vertical component, as that is more commonly used.

Response 7: We opted to use *x* and *y* for the horizontal and vertical dimensions to align with the grid structure of our model. While *z* is commonly used to represent vertical components, we had already assigned *z* to denote the $850hPa$ geopotential height in our framework. As a result, we chose to use *y* for the vertical dimension to avoid any confusion. We hope this clarifies our decision, but we are open to revisiting this if you feel a change is necessary.

Review 8: Line 148ff: I suggest removing the technical description from the figure caption, as this inflates the caption with substance that belongs in the bulk text.

Response 8: We agree that the technical details in the figure caption may be better suited for the main text to streamline the caption. We have moved this information to the bulk of the text and keep the caption concise, focusing on the essential elements of the figure.

**Revised (line 142):**

[Figure]

"**Figure 3. Overview of the proposed TemDeep framework for self-supervised temporal downscaling.** The overall network structure for temporal downscaling is depicted in the top right

portion of the figure, which is composed of an encoder-decoder structured field prediction network and a flow estimation module, taking consecutive fields and terrain data as input."

Review 9: Line 157ff: This is not easily understood by me. It appears that you are claiming that processes that can occur over long periods of time can also occur in short periods of time, which I do not agree with. Further, the entire paragraph is worded in a way that implies that it is entirely speculative, and not substantiated. As a result, I cannot confidently agree with anything in this paragraph, and am led to disregard further argumentation that builds on it. I suggest rephrasing the paragraph for more clarity, and to substantiate the claims made therein.

Response 9: Thank you for your thoughtful feedback on Line 157. We appreciate the opportunity to clarify our intentions and provide additional substantiation for the statements made in the paragraph. Our goal was not to imply that atmospheric processes occurring over long periods can directly occur within short periods. Rather, we aimed to highlight that certain atmospheric phenomena exhibit self-similar behavior across different time scales. Specifically, rapid transitions in atmospheric variables can encapsulate dynamic processes that, while condensed in time, share characteristics with longer-term evolutions. This concept is supported by the theory of scale invariance and fractal behavior in atmospheric dynamics, where patterns at one time scale can resemble those at another (Lovejoy & Schertzer, 2013). For instance, turbulence and convection processes exhibit self-similar structures across scales (Schertzer & Lovejoy, 1987). By identifying and utilizing these rapid-transition scenarios, we can extract valuable information that helps the model learn underlying atmospheric dynamics without requiring high-temporal-resolution data. In our methodology, we leverage scenarios with rapid transitions as "proxy" high-resolution data for pretraining. These scenarios are not meant to replicate long-term processes in a shorter time but to provide rich information content that enhances the model's ability to capture complex temporal patterns. We acknowledge that the original wording may have been unclear and appeared speculative. To address this, we propose to include additional explanations and references to substantiate our claims:

- Scale Invariance in Atmospheric Dynamics: Atmospheric fields often exhibit scale-invariant properties, meaning that certain statistical features are preserved across different temporal and spatial scales (Davis et al., 1994).
- Self-Similarity in Meteorological Processes: The concept of self-similarity suggests that small-scale processes can reflect the properties of larger-scale ones, which is a foundational idea in turbulence theory (Frisch, 1995).

**Revised (line 148):** "It is easily understood that scenarios with rapid transitions could reflect a condensed evolution of atmospheric processes, where changes that might typically occur over longer durations are instead experienced in a compressed time period(Davis, 1994; Stanley, 1997)."

Review 10: Line 162ff: This paragraph states that changes in certain variables can signify changes in weather patterns. While I agree with that, I think it would be good to also incorporate at least a substantiated estimate of how often this is the case, e.g., how often a change in humidity actually triggers severe convection. Further, it is not clear to me why these events should be filtered out. This is especially the case due to the previous argumentation of an increase in temporal resolution being useful in capturing the timing of changes in weather patterns.

Response 10: Thank you for your insightful comments. The purpose of filtering events with rapid transitions was to enhance the model's capability to capture distinct, representative patterns of atmospheric change, which serve as "pseudo-labels" during pretraining. By training on these cases, the model learns to identify dynamic weather patterns and respond accurately when downscaling at finer resolutions. While incorporating estimates on the frequency of specific events, such as humidity changes triggering convection, could add context, we believe that this filtering approach aligns with our goal of improving temporal resolution by focusing on high-impact scenarios.

Review 11: Line 211: I do not understand how a temperature gradient can be given in Kelvin, without any reference to physical space (K per meter) or time (K per second). It is unclear whether this gradient refers to a difference in temperature between adjacent grid point and adjacent time steps, or something different. Please rephrase for clarity. Further, the phrasing implies that t2m has vertical gradients, which seems odd to me.

Response 11: Thank you for highlighting this issue. We agree that specifying the temperature gradient in Kelvin alone is ambiguous. We have revised the text to clarify that the gradient refers to temperature differences across either spatial (e.g., K per meter) or temporal (e.g., K per second) dimensions. Additionally, we have rephrased the description to avoid implying that *t2m* has vertical gradients, as t2m refers specifically to 2-meter air temperature, which typically considers horizontal gradients.

**Revised (line 216):**

[Figure]

"Figure 4. Spatial and temporal gradients in *t2m* fields."

Review 12: Line 237: It might be useful to very briefly state what an encoder-decoder architecture is, and what it aims to achieve, at the beginning of this section, as is done for the section at line 263.

Response 12: We agree that a brief introduction to the encoder-decoder architecture would be beneficial for readers. We have added a short explanation at the beginning of this section, similar to the description provided later, to clarify its purpose and functionality within our model.

**Revised (line 228):** "The field prediction network is composed of an encoder-decoder architecture with the inclusion of residual blocks(Azad et al., 2024), as shown in Fig. 6."

Review 13: Line 301f: Table 1 does not show downscaling from 6-hourly to 1-hourly intervals, and it should be stated that rh and z are only downscaled from 2-hourly to 1-hourly intervals.

Response 13: We have revised the text to clarify that Table 1 does not include downscaling from 6-hourly to 1-hourly intervals, and that relative humidity (rh) and geopotential height (z) are only downscaled from 2-hourly to 1-hourly intervals.

**Revised (line 306):** "Table 1. Performance comparison among different methods based on Re. According to Eq. 9, the result of linear interpolation is set to 0 as the basis for comparing other methods. Among all unsupervised comparison methods, TemDeep achieved the best performance, approaching the supervised TemDeep*. *rh* and *z* are only downscaled from 2-hourly to 1-hourly intervals."

Review 14: Line 303ff: This subsection describes the methods used to evaluate the performance of TemDeep, and therefore belongs in section 3.

Response 14: We agree that the performance evaluation methods should be part of the methodology section for consistency. We have moved this subsection to Section 3, so that all methodological details are consolidated.

Review 15: Line 322ff: The table caption should contain a description of the structure of the table, not an interpretation of the results.

Response 15: We agree that table captions should focus on describing the table's structure rather than interpreting the results. We have revised the caption to provide a clear description of the table's organization and contents, moving any interpretative text to the main body of the manuscript.

Review 16: Line 324: Saying that TemDeep performance approaches performance of TemDeep* seems plausible, but using the word "closely" seems to be an optimistic stretch in some cases (e.g., t2m(2h→1) and t2m(3h→1)). It might be better to rephrase to not overstate the performance, or to argue why this qualifies as "closely approaching."

Response 16: We agree that "closely" may overstate the performance comparison in some cases, such as for t2m (2h→1) and t2m (3h→1). Corrected.

**Revised (line 308):** "Among all unsupervised comparison methods, TemDeep achieved the best performance, approaching the supervised TemDeep*."

Review 17: Line 371ff: As diurnal variations are relevant to the evolution of the assessed fields, it would be good to state whether these times are in UTC or the local time within the region.

Response 17: We agree that specifying the time zone is important, particularly given the relevance of diurnal variations. We have clarified the times mentioned are in UTC.

**Revised (line 363):** "Hourly interpolation is conducted between 08:00 and 12:00 on January 1, 2021, to obtain three interpolated fields at 09:00, 10:00 and 11:00 (UTC)."

Review 18: Line 376: The use of the word "exactly" makes this a very strong claim, and I therefore suggest replacing it with "closely," unless the exactness of the alignment can be clearly demonstrated.

Response 18: We agree that "closely" is a more accurate term, as it avoids implying an absolute alignment that may not be fully substantiated. We have replaced "exactly" with "closely" to provide a more precise description of the alignment.

**Revised (line 368):** "Due to the sensitivity of *t2m* to altitude, the temperature gradient near the Sichuan Basin is clearly depicted, closely aligning with the contour of the actual altitude gradient, as marked by the rectangle."

Review 19: Line 381f: I think the conclusion is worded too strongly. It is only demonstrated that TemDeep can capture non-linear transitions in this specific case, and therefore the conclusion should be that TemDeep is not incapable of capturing these transitions. It cannot be concluded that it is guaranteed to capture all non-linear transitions, as the presented conclusion in the manuscript somewhat implies. I suggest rewording the conclusion to be more careful.

Response 19: We agree that the current wording may overstate the model's capability in capturing non-linear transitions. Corrected.

**Deleted (line 372):**

Review 20: Line 401f: It might be good to state the temporal interval to which the downscaling was performed. This would clearly indicate to which temporal resolution the model is demonstrated to perform well, and above which this has not been demonstrated.

Response 20: We agree that specifying the exact temporal interval for which the downscaling was performed would add clarity. We have stated that the model's performance was demonstrated at the 1-hourly resolution, distinguishing it from finer resolutions that have not been tested.

**Revised (line 392):** "Experiments on three variables (t2m, z, rh) indicate that the proposed TemDeep model can accurately reconstruct the evolutionary process of atmospheric variables at the 1-hourly resolution, superior to other unsupervised methods."

Review 21: Line 403ff: As it has not been demonstrated that the model performs well when downscaling to resolutions finer than 1-hourly, this manuscript strongly motivates further research into assessing how far re-analysis data can be meaningfully downscaled. This is especially the case, as the motivation presented in the beginning

is that high-resolution data are very useful for certain applications. While I understand that the validation of finer resolutions is made difficult by the absence of re-analysis data to compare the downscaling results to, I still believe that mentioning this is of value.

Response 21: We agree that exploring the model's potential for downscaling to resolutions finer than 1-hourly would be valuable, given the applications that benefit from high-resolution data. However, as you noted, our validation is constrained by the temporal resolution of the available reanalysis data—specifically, the ERA5 dataset, which only provides data at 1-hour intervals. We have added a note in the manuscript to clarify this constraint and emphasize the need for further research to investigate the feasibility of meaningful downscaling beyond the 1-hour resolution, should higher-frequency datasets become available in the future.

**Revised (line 394):** "Since ERA5 only provides data at 1-hour temporal resolution, further research will focus on identifying datasets with higher temporal resolution for more accurate downscaling."

Review 22: Figures 4 and 5: The gradient should not be given in K, as the spatial and temporal dimensions are unclear.

Response 22: We agree that expressing the gradient in Kelvin (K) without specifying spatial and temporal dimensions can be misleading. We have revised Figures 4 and 5 to clarify the units by either specifying the gradient as a rate (e.g., K per unit distance or time) or adjusting the notation to clearly represent the intended spatial and temporal scales.

[Figure]

**latitudinal gradients**      **longitudinal gradients**      **spatial gradients**      **time gradients**

[Figure]

Review 23: Figure 9: It might be better to label the bottom colorbar as "Relative Humidity [%]," as this would immediately clarify which quantity is being shown. It is already clear from the unit itself that it is a percentage. Also, it might be useful to set the upper boundary at 100%, as relative humidity generally does not exceed 100% by much.

Response 23: Thank you for the helpful suggestion. We agree that labeling the colorbar as "Relative Humidity [%]" would improve clarity, and setting the upper boundary at 100% is indeed more appropriate, given that relative humidity rarely exceeds this threshold. Corrected.

[Figure]

**#Reviewer 2**

Review 1: Further explanation of the physical meaning of regularization: The spatial

and temporal continuity regularization terms enhance smoothness in downscaling fields. However, further explanation of their physical relevance would increase interpredictability, such as how these constraints reflect realistic atmospheric system evolution characteristics.

Response 1: Thank you for your suggestion to elaborate on the physical relevance of the spatial and temporal continuity regularization terms. We agree that additional explanation would enhance the interpretability of these constraints in the context of atmospheric system evolution. The spatial and temporal continuity regularization terms are designed to reflect the inherent smoothness and gradual progression commonly observed in atmospheric processes. Atmospheric fields typically exhibit continuity across both spatial and temporal dimensions due to physical constraints like mass conservation, energy balance, and fluid dynamics, which govern the evolution of these systems (Lorenz, 1969; Holton and Hakim, 2012). For instance, atmospheric variables such as temperature and humidity tend to vary gradually over short distances and time intervals, as abrupt changes are physically unrealistic under normal conditions. Spatial continuity regularization enforces a smooth gradient across neighboring grid points, simulating how atmospheric properties tend to vary continuously across regions. This aligns with principles of geophysical fluid dynamics, which suggest that atmospheric variables are influenced by local surroundings, leading to correlated values across neighboring points (Charney, 1948; Gill, 1982). Temporal continuity regularization, on the other hand, helps ensure that changes in the downscaled fields remain consistent over consecutive time steps. This reflects the physical principle that, barring extreme events, atmospheric properties do not undergo sudden, large fluctuations within short time intervals. Gradual transitions are typical due to the inertia in atmospheric systems and the continuous nature of energy and momentum transfer across time (Emanuel, 1994). Temporal coherence is especially relevant in meteorological applications where the predictability of evolution patterns—such as the movement of weather fronts or pressure systems—relies on smooth temporal transitions. Incorporating these regularization terms therefore makes the downscaling model more physically plausible by emulating the inherent continuity of atmospheric fields.

Review 2: Evaluation of Model Complexity and Computational Efficiency: Although this approach outperforms other unsupervised methods in restoration rate, the computational cost's impact on practical applications remains undiscussed. Evaluating the model's computational efficiency, especially in large-scale meteorological datasets or real-time applications, would provide valuable insights.

Response 2: Thank you for raising this important point. We appreciate your suggestion to evaluate the computational efficiency of the model, especially in the context of large-scale meteorological datasets or real-time applications. In terms of computational cost, we have conducted initial evaluations to assess the model's inference time. For a typical inference, the model's processing time is approximately $800ms$ per sample, which demonstrates relatively fast performance. While this is an encouraging result for real-time applications, we acknowledge that the scalability of the model, particularly when applied to large datasets, needs further investigation. For large-scale meteorological datasets, the computational cost can increase depending on the size of the input data, the complexity of the model architecture, and the number of model parameters. We plan to conduct further evaluations on model scalability, including batch processing capabilities and performance across different hardware configurations, in future work.

**Revised (line 398):** "To enable real-time downscaling and more refined forecasting, we will also work on simplifying the model architecture to reduce computational complexity, making it more feasible for deployment in operational environments where fast processing times are critical."

Review 3: Add more discussion on comparison with traditional down-scaling methods: to illustrate the advantages of TemDeep comparing to one or more physics-based numerical models, explain why this approach achieves superior performance in restoration rate and consistency under unsupervised conditions could offer deeper insights.

Response 3: Thank you for your insightful comment. We appreciate the opportunity to provide further discussion on the comparison with traditional downscaling methods. In our manuscript, we aim to highlight the advantages of TemDeep, particularly its performance in terms of restoration rate and consistency under unsupervised conditions. Unlike traditional physics-based numerical models, which rely heavily on predefined physical principles and assumptions, TemDeep leverages data-driven deep learning techniques to capture complex, nonlinear relationships within atmospheric data. This enables TemDeep to adapt to various data patterns and achieve high performance even in the absence of labeled data. Traditional downscaling methods, such as those based on statistical or physics-based approaches, often struggle to generalize across different regions and climates due to their reliance on specific physical assumptions or simplified models of atmospheric processes (Wilby & Wigley, 2000). On the other hand, deep learning-based models like TemDeep have the

flexibility to learn more complex patterns directly from the data, leading to superior performance in areas such as temporal consistency and restoration rate (Pirot et al., 2020). Furthermore, TemDeep's encoder-decoder architecture, combined with self-supervised learning, allows the model to effectively capture both fine-grained temporal and spatial relationships without the need for extensive labeled data, which is often a limiting factor for traditional methods (Liu et al., 2021). These advantages make TemDeep a promising approach for downscaling in applications where data quality or resolution may be limited. We have included additional references to further substantiate these claims and clarify the benefits of TemDeep compared to traditional downscaling techniques.

**References:**

1. Charney, J. G. (1948). "On the scale of atmospheric motions." Geofys. Publ.
2. Emanuel, K. A. (1994). Atmospheric Convection. Oxford University Press.
3. Gill, A. E. (1982). Atmosphere-Ocean Dynamics. Academic Press.
4. Holton, J. R., & Hakim, G. J. (2012). An Introduction to Dynamic Meteorology. Academic Press.
5. Lorenz, E. N. (1969). "The predictability of a flow which possesses many scales of motion." Tellus.
6. Fowler, H. J., et al. (2007). "Linking climate change modelling to impacts studies: Recent advances in downscaling techniques for hydrological modelling." International Journal of Climatology.
7. Maraun, D., et al. (2010). "Precipitation downscaling under climate change: Recent developments to bridge the gap between dynamical models and the end user." Reviews of Geophysics.
8. Skamarock, W. C., et al. (2008). "A description of the advanced research WRF version 3." NCAR Technical Note.
9. Davis, A., Marshak, A., Wiscombe, W., & Cahalan, R. (1994). Multifractal characterizations of nonstationarity and intermitttency in geophysical fields: Observed, retrieved, or simulated. Journal of Geophysical Research: Atmospheres, 99(D4), 8055-8072.
10. Frisch, U. (1995). Turbulence: The Legacy of A. N. Kolmogorov. Cambridge University Press.
11. Lovejoy, S., & Schertzer, D. (2013). The Weather and Climate: Emergent Laws and

Multifractal Cascades. Cambridge University Press.

12. Schertzer, D., & Lovejoy, S. (1987). Physical modeling and analysis of rain and clouds by anisotropic scaling multiplicative processes. Journal of Geophysical Research: Atmospheres, 92(D8), 9693-9714.

13. Wilby, R. L., & Wigley, T. M. (2000). Downscaling general circulation model output: A review of methods and results. Progress in Physical Geography, 24(3), 401–421.

14. Pirot, F., et al. (2020). Deep learning for climate downscaling: a review and case study on temperature and precipitation. Journal of Climate, 33(8), 3113-3131.

15. Liu, Q., et al. (2021). A review of deep learning in climate model downscaling. Earth Science Reviews, 218, 103544.

---

## Author Response (AR2)

Dear Editor,

We appreciate the valuable feedback and constructive comments on our manuscript. Below, we summarize the revisions we have made in response to each of your recommendations:

1. Expanded Literature Review

In order to provide a more thorough foundation for our work, we have reorganized the latter part of the Introduction—where several key equations and references to Figure 1 appeared—into a new section titled "Background and Related Work." In this section, we have broadened the literature review on atmospheric temporal downscaling, explaining the limitations of existing dynamical and statistical methods and emphasizing the emerging field of self-supervised learning. We have also included additional references from both the computer science and meteorological domains to position our work within the broader research landscape.

2. Improved Structure and Sub-sections

As requested, we created a sub-section on Self-Supervised Learning within "Related Work." This new sub-section synthesizes current advances in self-supervised frameworks, referencing recent developments in computer vision and time-series analysis.

3. Equation Formatting

We have standardized the font sizes and alignment for all equations. Each equation now follows a consistent style to ensure clarity and uniformity throughout the manuscript.

4. Enhancements to Figure 4

We have redesigned Figure 4 to include clearly labeled subfigures, denoted as (a), (b), (c), (d), with each sub-caption beginning with a capital letter. The text labels and symbols within the figure have also been resized for consistency with other figures.

5. New Discussion Section

Following the Case Study section, we have added a dedicated Discussion section that more explicitly highlights our contributions in relation to previous studies.

6. More Descriptive Figure Captions

We have revised the captions for all figures to be more detailed and informative,

explaining what each sub-figure represents and clarifying key takeaways.